# Copilot Arena: A Platform for Code LLM Evaluation in the Wild

Wayne Chi [* 1]   Valerie Chen [* 1]   Anastasios Nikolas Angelopoulos [2]   Wei-Lin Chiang [2]   Aditya Mittal [1]
Naman Jain [2]   Tianjun Zhang [2]   Ion Stoica [2]   Chris Donahue [† 1]   Ameet Talwalkar [† 1]

## Abstract

Evaluating in-the-wild coding capabilities of large language models (LLMs) is a challenging endeavor with no existing solution. We introduce Copilot Arena, a platform to collect user preferences through native integration into a developer's working environment. Copilot Arena comprises a novel interface for comparing pairs of model outputs, a sampling strategy to reduce experienced latency, and a prompting scheme to enable code completion functionality. Copilot Arena has served over 4.5 million suggestions from 10 models and collected over 11k pairwise judgements. Our results highlight the importance of model evaluations in integrated settings. We find that model rankings from Copilot Arena differ from those of existing evaluations, which we attribute to the unique distribution of data and tasks contained in Copilot Arena. We also identify novel insights into human preferences on code such as an observed consistency in user preference across programming languages yet significant variation in preference due to task category. We open-source Copilot Arena and release data to enable human-centric evaluations and improve understanding of coding assistants.

## 1. Introduction

As model capabilities improve, large language models (LLMs) are increasingly integrated into user environments and workflows. For example, software developers code with AI in integrated developer environments (IDEs) (Peng et al., 2023), doctors rely on notes generated through ambient listening (Oberst et al., 2024), and lawyers consider case

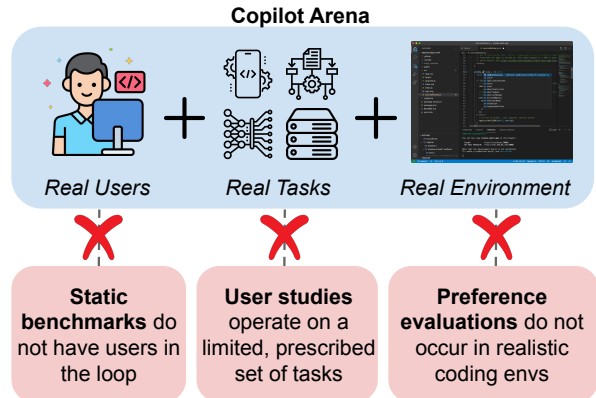

**Copilot Arena**

*Real Users* + *Real Tasks* + *Real Environment*

| Static **benchmarks** do not have users in the loop | **User studies** operate on a limited, prescribed set of tasks | Preference **evaluations** do not occur in realistic coding envs |

**Limitations of existing evaluations**

*Figure 1.* Copilot Arena is a platform for conducting realistic evaluations of code LLMs, collecting human preferences of coding models with real users, real tasks, and in realistic environments, aimed at addressing the limitations of existing evaluations.

evidence identified by electronic discovery systems (Yang et al., 2024a). Increasing deployment of models in productivity tools demands evaluation that more closely reflects real-world circumstances (Hutchinson et al., 2022; Saxon et al., 2024; Kapoor et al., 2024). While newer benchmarks and live platforms incorporate human feedback to capture real-world usage, they almost exclusively focus on evaluating LLMs in chat conversations (Zheng et al., 2023; Dubois et al., 2023; Chiang et al., 2024; Kirk et al., 2024). Model evaluation must move beyond chat-based interactions and into specialized user environments.

In this work, we focus on evaluating LLM-based coding assistants, specifically focusing on their ability to generate code completions. Despite the popularity of these tools—millions of developers use Github Copilot (Github, 2022)—existing evaluations of the coding capabilities of new models exhibit multiple limitations (Figure 1, bottom). Traditional ML benchmarks evaluate LLM capabilities by measuring how well a model can complete static, interview-style coding tasks (Chen et al., 2021; Austin et al., 2021; Jain et al., 2024a; White et al., 2024) and lack *real users*. User studies recruit real users to evaluate the effectiveness of

---
[*]Equal contribution . [†]Co-senior Authors. [1]Carnegie Mellon University [2]UC Berkeley. Correspondence to: Wayne Chi <waynechi@andrew.cmu.edu>, Valerie Chen <valeriechen@cmu.edu>.

*Proceedings of the $42^{nd}$ International Conference on Machine Learning*, Vancouver, Canada. PMLR 267, 2025. Copyright 2025 by the author(s).

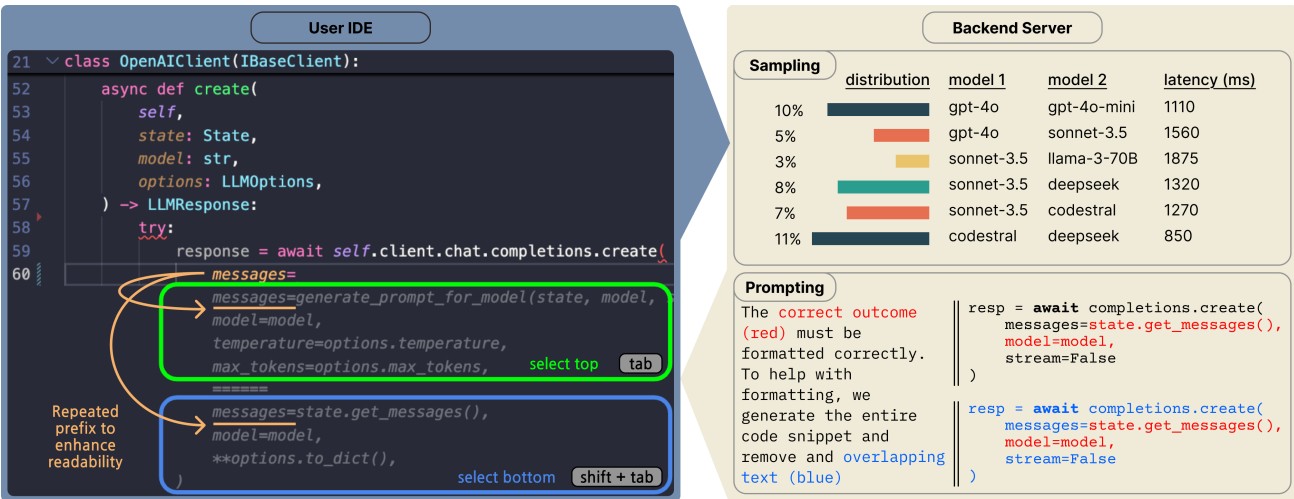

*Figure 2.* We introduce Copilot Arena, a VSCode extension to collect human preferences of code directly in a developer's IDE. Copilot Arena enables developers to use code completions from various models. The system comprises a) the interface in the user's IDE which presents paired completions to users (left), b) a sampling strategy that picks model pairs to reduce latency (right, top), and c) a prompting scheme that allows diverse LLMs to perform code completions with high fidelity. Users can select between the top completion (green box) using `tab` or the bottom completion (blue box) using `shift+tab`.

LLMs as coding assistants, but are often limited to simple programming tasks (Vaithilingam et al., 2022; Ross et al., 2023; Mozannar et al., 2024b) as opposed to *real tasks*. Recent efforts to collect human feedback such as Chatbot Arena (Chiang et al., 2024) are still removed from a *realistic environment*, resulting in users and data that deviate from typical software development processes. We introduce Copilot Arena to address these limitations (Figure 1, top), and we describe our three main contributions below.

**We deploy Copilot Arena in-the-wild to collect human preferences on code.** Copilot Arena is a Visual Studio Code extension, collecting preferences directly in a developer's IDE within their actual workflow (Figure 2). Copilot Arena provides developers with code completions, akin to the type of support provided by Github Copilot (Github, 2022). Over the past 3 months, Copilot Arena has served over 4.5 million suggestions from 10 state-of-the-art LLMs, gathering 11604 votes across 1642 users. To collect user preferences, Copilot Arena presents a novel interface that shows users paired code completions from two different LLMs, which are determined based on a sampling strategy that aims to mitigate latency while preserving coverage across model comparisons. Additionally, we devise a prompting scheme that allows a diverse set of models to perform code completions with high fidelity.

**We construct a leaderboard of user preferences and find notable differences from existing static benchmarks and human preference leaderboards.** In general, we observe that smaller models seem to overperform in other static benchmarks compared to our leaderboard, while performance among larger models is mixed. We attribute these differences to the fact that Copilot Arena is exposed to users and tasks that differ drastically from code evaluations in the past. Our data spans 103 programming languages and 19 natural languages as well as a variety of real-world applications and code structures, while static benchmarks tend to focus on a specific programming and natural language and task (e.g. coding competition problems). Additionally, while all of Copilot Arena interactions contain code contexts and the majority involve infilling tasks, a much smaller fraction of Chatbot Arena's coding tasks contain code context, with infilling tasks appearing even more rarely.

**We derive new insights into user preferences of code by analyzing Copilot Arena's diverse and distinct data distribution.** We compare user preferences across different stratifications of input data (e.g., common versus rare languages) and observe which affect observed preferences most. For example, while user preferences stay relatively consistent across various programming languages, they differ drastically between different task categories (e.g. frontend/backend versus algorithm design). We also observe variations in user preference due to different features related to code structure (e.g., context length and completion patterns). We open-source Copilot Arena, all user votes, and a curated subset of code contexts.[1] Altogether, our results highlight the necessity of model evaluation in realistic and

---

[1]All code and data is available at `https://github.com/lmarena/copilot-arena`.

domain-specific settings.

## 2. System Design

Copilot Arena is a VSCode extension that provides users with pairs of inline code completions from various LLMs. In return, users provide their votes on which completion is better suited for their task. To avoid interrupting user workflows, voting is designed to be *seamless*—users use keyboard shortcuts to quickly accept one of the two completions into their code, which we interpret as a vote in favor of the underlying model that produced it. Designed to allow for developer's day-to-day usage, the three core components of Copilot Arena—overviewed in Figure 2—are the 1) User Interface, 2) Model Sampling, and 3) Model Prompting.

### 2.1. User Interface

Traditional code completion tools (e.g., GitHub Copilot (Github, 2022)) only show one completion at a time. However, showing two code completions simultaneously enables us to collect preference judgments on the same context and facilitate a more seamless user experience (Chiang et al., 2024; Lu et al., 2024). As such, we propose an interface that allows a user to view two completions in a head-to-head manner; to our knowledge, we are the first to introduce an interface that does so. We propose a design inspired by Git Diff—a well-established tool familiar to many developers—which displays code from the current commit and code from the incoming commit stacked vertically, one on top of the other. In a similar manner, given an existing code context, we also stack responses from two different model outputs. This allows users to examine both completions together (an example of how the completions are visualized is in Figure 2). The user can accept the top suggestion using `tab` and the bottom suggestion using `shift+tab`, or decide neither is appropriate and continue typing. The only distinction between our system and conventional inline completion systems is the inclusion of a second suggestion, resulting in a user experience that is familiar overall.

We make several other notable design decisions. First, we repeat the first line in the ghost text of the top completion so that both top and bottom completions are entirely ghost text. Not repeating the text—as is the case with a single completion—was an alternative we considered, but our initial pilot studies indicated that the discrepancy between top (partial ghost text) and bottom (full ghost text) completions was more likely to confuse users. Second, we always wait for both completions to finish generating before showing them to the user to reduce the effects of latency on user preference, which we aim to study separately in Section 5.2. Lastly, we randomize the ordering of the completions to remove top-bottom bias from our preference evaluation. We discuss additional design decisions in Appendix A.1.

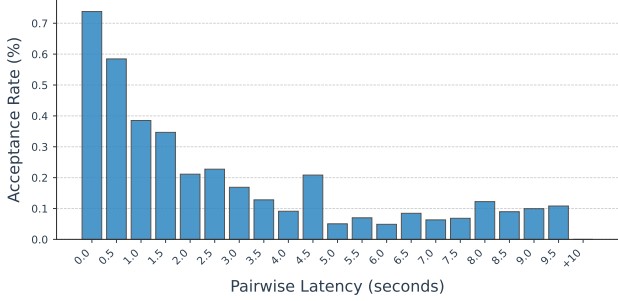

*Figure 3.* The likelihood of users accepting one of the two completions as a function of empirical pairwise latency (determined by the slower of the two models). As latency increases, users are less likely to accept a completion. We devise a sampling strategy described in Section 2.2 which reduces pairwise latency by 33% while also ensuring sufficient coverage of unique model pairs.

### 2.2. Model Sampling

A key challenge in building a realistic environment for coding assistance is providing responsive code completion. Developer expectations for low latencies impact not only user satisfaction and retention, but also directly affect their likelihood to provide preference data. The slower the completions are returned to the user, the *less likely* users are to vote (i.e. users select neither completion) (Figure 3). However, many model providers do not optimize their API endpoints for low-latency use cases, requiring us to explore a sampling strategy that improves our system-wide latency.

Since the Copilot Arena interface shows two code completions together, the slowest completion determines the latency. Thus, given a set of $M$ models $\{1, \ldots, M\}$, we let $F_{\max}(l; i, j)$ denote the cumulative density function (CDF) for the maximum latency between models $i$ and $j$. Because latencies tend to be long-tailed, we model $F_{\max}(l; i, j)$ as a log-normal CDF with parameters estimated from our historical data. Our objective will then be to minimize the expected latency of the chosen model pair under the distribution induced by our observed data,

$$\mathcal{L}(\theta) = \mathbb{E}_{(i,j) \sim p_\theta, L \sim F_{\max}(l; i, j)}[L], \qquad (1)$$

where $p_\theta$ is a distribution over model pairs,

$$p_\theta(i, j) = \frac{\exp(\theta_{ij}/\tau)}{\sum_{k<l} \exp(\theta_{kl}/\tau)}. \qquad (2)$$

Above, $\tau$ is a temperature parameter that interpolates between a latency-optimized distribution and a uniform distribution, allowing us to trade off latency and coverage of unique model pairs. The parameters $\theta \in \mathbb{R}^{\binom{M}{2}}$ are optimized via gradient descent to minimize (Eq. 1). In practice, we set $\tau$ to values between 5 and 10 to ensure sufficient coverage. By deploying our algorithm, we observed a decrease

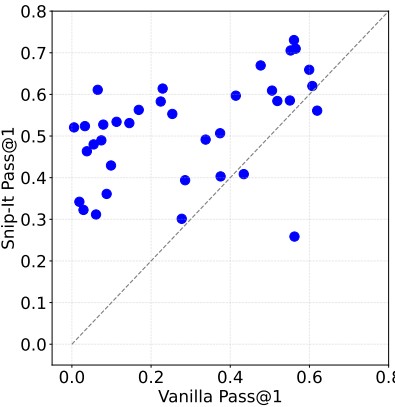

*Figure 4.* We evaluate the effectiveness of our Snip-It method by comparing LLM performance on infilling tasks (using pass@1) before and after applying Snip-It. We evaluate 9 different models of varying performance across 4 different prompt templates (i.e., ways of encoding the prefix and suffix in the prompt): each point represents one model and one prompt template pair. We observe that, across the board, all pairs generally benefit from the Snip-It method (e.g., lie above the diagonal line).

in median experienced latency by 33% (from 1.61 to 1.07 seconds) compared to a uniform distribution.

## 2.3. Model Prompting

During real development processes, developers frequently modify or expand upon existing code which requires models to *infill* between code segments. However, many popular coding models such as GPT-4o or Sonnet 3.5 are instruction-tuned (Wei et al., 2022) and trained to output text left-to-right autoregressively, rather than to "fill-in-the-middle" (FiM) (Fried et al., 2023; Gong et al., 2024). In preliminary experiments, we observed poor, essentially unusable performance of instruction-tuned models on FiM tasks. Accordingly, we use offline datasets to improve chat models' infilling capabilities through prompt engineering. We include full experimental details and results in Appendix A.2.

**Offline Evaluation Set-up.** Our set-up uses the HumanEval-infilling dataset (Bavarian et al., 2022) which consists of 1640 examples where random spans in a completed portion of code are masked out to simulate FiM behavior. To incorporate prefix and suffix information, we began with several prompt templates from Gong et al. (2024) with modifications to align the prompts with chat models (e.g., initial instruction and few-shot examples). The templates capture different ways to encode information about the given code context. For example, prefix-suffix-middle presents the code context in the order of prefix and then suffix, and the LLM is asked to output the middle.

**Vanilla performance on FiM tasks.** We find that the suc-

cess of standard prompt templates varies greatly between models (Table 2). This is not necessarily an indication that models cannot code as clearly many state-of-the-art chat models are proficient coders (Jain et al., 2024a; Lin et al., 2024). Instead, the vast majority of the errors result in formatting issues or duplicate code segments rather than logical errors, indicating that FiM performance is inhibited more by low-level formatting issues than high-level coding capabilities: see examples of these errors in Appendix A.2.3.

**Post-processing using Snip-It.** While it is not feasible to retrain these models because many of them offer API access only, we explore alternative approaches via prompting to improve chat models' abilities to complete FiM tasks. Specifically, we allow the model to generate code snippets, which is a more natural format, and then post-process the snippets into a FiM completion. Our approach—Snip-It—is as follows: the model is prompted with the same prompts as above (e.g. prefix-suffix-middle) but with instructions to begin by repeating a portion of the prefix and similarly end by repeating a portion of the suffix. Then, we remove any portion of the output code that already exists in the input, similar to recent agentic search-replace tools (Anthropic, 2024). As shown in Figure 4, we found that, relative to the baseline, Snip-It provides robust performance gains for infilling: performance improved in 93% of the conditions. High-performing models improve substantially (e.g., Claude-3.5-Sonnet improves from 56.1% to 73.0%), while initially struggling models improve dramatically (e.g., Llama-3.1-70B from 7.4% to 49%). While offline evaluation is not a perfect metric, we find that these drastic improvements enable these models for FiM tasks.

## 3. System Deployment

**Deployment Details.** The Copilot Arena extension is advertised in online open-source communities and made available on the VSCode extension store, where it is free to download. Similar to the set-up employed by Chiang et al. (2024) and Lu et al. (2024), participants are not compensated for using the extension, as in a traditional user study, but instead receive free access to state-of-the-art models. In addition to logging all preference judgments made by users of Copilot Arena, we also log the latency of each model response, the type of file the user is writing, the prefix and suffix length (characters and tokens), each completion length, which model was in the top versus bottom position, and a unique userID—all of which allows users to utilize the extension without revealing the content of what the user is working on. Given the sensitive nature of programming, we established clear privacy controls to give users the ability to restrict our access to their data. Depending on privacy settings, we also collect the user's code context and model responses. Appendix B provides a copy of the specific user

instructions and privacy guidelines. The data collection process is fully compliant with appropriate laws and regulations regarding user privacy within our institution and locale.

**Data collection process.** We select 10 state-of-the-art models to balance a set of open and commercial models, as well as generalist and code-specific models. In latter analysis, we refer to LLMs by shortened names to conserve space: please check Table 3 for full model names. Across 1642 users, we have served over 4.5 million suggestions and collected 11604 votes over the course of 3 months. Overall, we find that all models received between 2-5K votes, satisfying our coverage objective. In general, the median time to vote—the time taken after the completion is displayed to the user—was 7 seconds, suggesting that users did not accept all suggestions immediately and considered both completions. A more in-depth overview of data analysis is in Appendix C.

**Data Release.** Upon publication, we will release the full dataset of 11604 votes, with non-deanonymizing features (e.g., context length or file type). Despite giving users full control over their privacy, we take a conservative approach to data release given the potential sensitivity of coding data. To demonstrate the type of code users write using Copilot Arena, we will also release a hand-curated set of 500 examples that contain the prefix, suffix, and both completions. This portion of the dataset spans all 10 models and captures a variety of downstream applications—Appendix C shows multiple examples. Two authors carefully checked this set of examples to ensure the code also contained no sensitive information or personally identifiable information. We intend to continue a slow release in the future.

## 4. Model Rankings

### 4.1. Copilot Arena Leaderboard

We construct a leaderboard using our user preference judgements. Let $n$ denote the number of judgments and $M$ the number of models. For each battle $i \in [n]$, we define: $X_i \in \{-1, 0, 1\}$: $X_{i,m} = 1$ if model $m$ is presented in the top position, $X_{i,m} = -1$ if presented in the bottom position, and 0 otherwise. The outcome $Y_i \in \{0, 1\}$, where 1 indicates the top model won. As is standard in other work on pairwise preference evaluation (Chiang et al., 2024; Lu et al., 2024), we apply a Bradley-Terry (BT) model (Bradley & Terry, 1952) to estimate the relative strengths of models $\beta \in \mathbb{R}^M$, where the probability $p_{ij}$ that model $i$ beats model $j$ can be modeled as:

$$p_{ij} = \frac{e^{\beta_i}}{e^{\beta_i} + e^{\beta_j}}.$$

We bootstrap the battles in the BT calculation to construct a 95% confidence interval for the rankings, which are used to create a leaderboard that ranks all models, where each

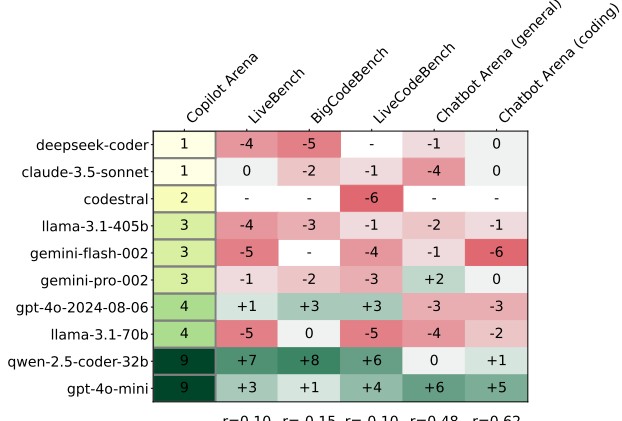

*Figure 5.* We compare model rankings in Copilot Arena (1st column) to existing evaluations, both static benchmarks (2nd-4th column) and live preference evaluations (last two columns). For existing evaluations, we show the *change* in rank relative to Copilot Arena rank, with positive values in green denoting models performing better on existing evaluations, negative values in red denoting models performing worse, and a dash indicating that the model is not present in the live leaderboard. We also report the Spearman rank correlation coefficients between Copilot Arena and other leaderboards.

model's rank is determined by which other models' lower bounds fall below its upper bound.

Constructing our leaderboard (Figure 5, 1st column), we find that our leaderboard is segmented into multiple tiers based on the estimated $\beta_i$ values (Table 4). In the first tier, DeepSeek Coder and Claude Sonnet-3.5 are at the top, with Codestral following closely behind. In general, we observe that code-specific models (e.g., DeepSeek Coder and Codestral) are competitive with general-purpose state-of-the-art models (e.g. Claude Sonnet-3.5), especially if they are trained to infill. In the second tier, there are 5 models of varying sizes and from different model providers that have relatively similar strengths. In the final tier, users preferred two models the least. In particular, Qwen-2.5-coder is an exception, performing notably worse than other code-specific models. Implementation of leaderboard computation and additional ablations on provided in Appendix D.

### 4.2. Comparison against Prior Evaluations

We compare our leaderboard to existing evaluations which encompass both live preference leaderboards with human feedback and static benchmarks (Figure 5, 2nd-5th column). For human preferences, we compare against Chatbot Arena (Chiang et al., 2024)—an in-the-wild platform for evaluating human preferences on chat capabilities—across both the general leaderboard and the coding subset. For static coding benchmarks, we select three that are recent and continue to be maintained (of which we have at

*Table 1.* We compare Copilot Arena with prior evaluations in terms of scale, context length, task type, and code structure. Copilot Arena provides broad coverage across programming languages (**PL**), natural languages (**NL**), **context length** in characters, **multiple task types**, and structural dimensions—whether the context contains **code** and fill-in-middle (**FiM**) tasks are present. Chatbot Arena (code) only contains code in 40% and infilling in 2.6% of its input and is denoted by ✗. In Figure 5, we compare against benchmarks that are updated with the latest models (denoted by *).

| | Scale | | Context Len | | Task | Structure | |
|---|---|---|---|---|---|---|---|
| **Benchmark** | # PL | # NL | p50 | p95 | Multi | Code | FiM |
| Copilot Arena | 103 | 19 | 1.6k | 18k | ✓ | ✓ | ✓ |
| HumanEval | 1 | 1 | 0.4k | 0.9k | ✗ | ✓ | ✗ |
| HumanEval-XL | 12 | 23 | 0.4k | 0.9k | ✗ | ✓ | ✗ |
| SAFIM | 4 | 1 | 3k | 5.9k | ✓ | ✓ | ✓ |
| LiveCodeBench* | 1 | 1 | 1.4k | 2.5k | ✗ | ✓ | ✗ |
| LiveBench* | 1 | 1 | 2.3k | 3.9k | ✓ | ✓ | ✗ |
| BigCodeBench* | 1 | 1 | 1.1k | 1.9k | ✓ | ✓ | ✗ |
| Chatbot Arena (general)* | ≥ 17 | ≥ 49 | 0.7k | 2.9k | ✓ | ✗ | ✗ |
| Chatbot Arena (code)* | ≥ 17 | ≥ 39 | 1.4k | 7.8k | ✓ | ✗ | ✗ |

least 8 out of 10 overlapping models): LiveBench (White et al., 2024)—a collection of benchmarks designed to mitigate test set contamination—LiveCodeBench (Jain et al., 2024a)—a code benchmark designed to mitigate test set contaimnation—and BigCodeBench (Zhuo et al., 2024)–a benchmark for code generation which focuses on function calls. We do not compare to rankings from any user studies because they are difficult to keep updated in comparison to both static benchmarks and live comparative systems.

We find the highest correlation ($r_s = 0.62$) with Chatbot Arena (coding) (Chiang et al., 2024) and similarly high correlation ($r_s = 0.48$) with Chatbot Arena (general). However, we find a low correlation ($r_s \leq 0.1$) with most static benchmarks. The stronger correlation with human preference evaluations compared to static benchmarks likely indicates that human feedback captures distinct aspects of model performance that static benchmarks fail to measure. We notice that smaller models tend to overperform (e.g., GPT-4o mini and Qwen-2.5-Coder 32B), particularly in static benchmarks. We attribute these differences to the unique distribution of data and tasks that Copilot Arena evaluates over, which we explore in more detail next.

## 5. Data Analysis

### 5.1. Exploring Copilot Arena Data

Evaluating models in real user workflows leads to a diverse data distribution in terms of programming and natural languages, tasks, and code structures—e.g., context lengths, last-line contexts, and completion structures (Figure 6). We discuss how our data distribution compares against those

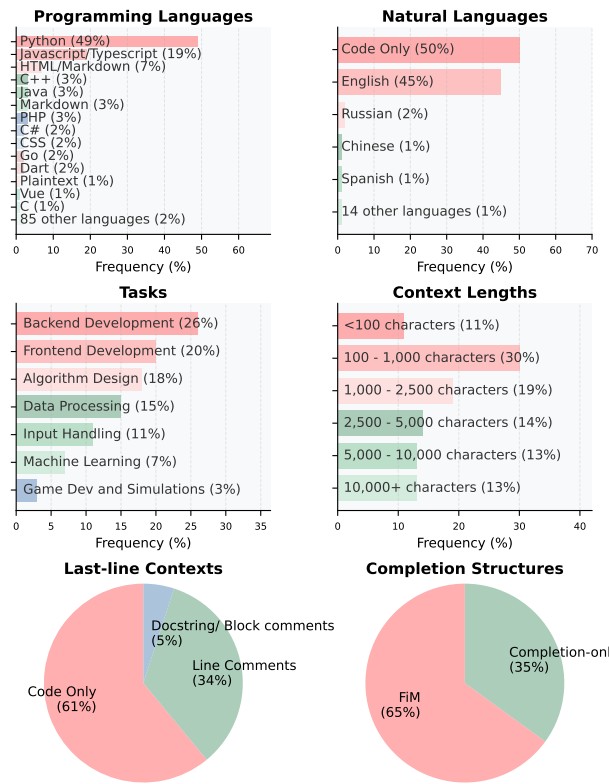

*Figure 6.* Copilot Arena data is diverse in programming and natural languages, downstream tasks, and code structures (e.g., context lengths, last-line contexts, and completion structures).

considered in prior evaluations (Table 1).

**Programming and Natural Language:** Previous benchmarks such as HumanEval (Chen et al., 2021) cover a limited number of languages, primarily focusing on Python and English (Bavarian et al., 2022; Jain et al., 2024a; White et al., 2024; Zhuo et al., 2024). While recent work such as HumanEval-XL (Peng et al., 2024) and SAFIM (Gong et al., 2024) has expanded coverage to up to a dozen programming languages, Copilot Arena covers 103 programming languages which is an order of magnitude more than most other benchmarks. While the distribution of programming languages is not uniform, there is still a core set of programming languages which contain a significant number of votes (Appendix C). Similarly, while the majority of Copilot Arena users (45%) write in English, we also identify 19 different natural languages which is comparable to Chatbot Arena (general) (Chiang et al., 2024) and benchmarks focused on multilingual generation (Peng et al., 2024).

**Downstream Tasks:** Existing benchmarks tend to source problems from coding competitions (Jain et al., 2024a; White et al., 2024), handwritten programming challenges (Chen et al., 2021), or from a curated set of GitHub repositories (Gong et al., 2024). In contrast, Copilot Arena

users are working on a diverse set of realistic tasks, including but not limited to frontend components, backend logic, and ML pipelines (we provide representative examples of the different task clusters in Appendix C.1). Coding style problems (i.e., algorithm design) comprise a much smaller portion—18%—of Copilot Arena's data. Further, the distribution of downstream tasks for our in-editor suggestions differs from questions raised by chat conversations, e.g., in Chatbot Arena (Chiang et al., 2024), where coding questions also focus on code explanation or suggesting commands.

**Code Structures and Context Lengths:** Most coding benchmarks follow specific structures, e.g., taking structured docstrings as input (Chen et al., 2021; Zhuo et al., 2024; Jain et al., 2024a; White et al., 2024) or infilling tasks (Bavarian et al., 2022; Gong et al., 2024). This means that most benchmarks have relatively short context lengths (e.g., all HumanEval (Chen et al., 2021) problems are less than 2k characters). Similarly, Chiang et al. (2024) focuses on natural language input collected from chat conversations, with many prompts not including any code context (e.g., 40% of Chatbot Arena's coding tasks contain code context and only 2.6% focus on infilling). As such, input prompts are also relatively short, with 95% of prompts falling between 1-3k characters. Unlike any existing evaluation, Copilot Arena is structurally diverse, comprising a mixture of infilling versus code completion and forms of docstring tasks. Since users are working in actual IDEs, they work on significantly longer inputs: the median context length is around 1.6k characters and 95% of inputs fall within 18k characters.

### 5.2. Understanding User Preferences of Code

Given our diversity of input features, we evaluate how each impacts user preference. We partition each feature into contrasting subsets (e.g. FiM vs non-FiM), which we refer to as $X$ and $\tilde{X}$. For each subset, we compute the win-rate[2] matrix $W \in \mathbb{R}^{M \times M}$ where $W(X)$ represents the win-rate matrix of subset $X$ and $W_{ij} \in [0, 1]$. For each feature, we compute a win-rate difference matrix $\Delta \in \mathbb{R}^{M \times M}$, which represents the number of substantial differences in the win-rate between $W(X)$ and $W(\tilde{X})$.

$$\Delta_{i,j} = \mathbb{1}[(W_{i,j}(X) - W_{i,j}(\tilde{X})) > \epsilon]$$

In our analysis, substantial changes are those in the top 90th percentile of win-rate changes ($\epsilon = 0.166$). Since $M = 10$, the maximum amount of significant changes is 90 ($|\Delta| \le 90$).

We compute $\Delta$ for four input features—task type, context length, FiM, and programming language—where contrasting strata are present in sufficient quantity ($\ge 10\%$) within

_______________

[2]Inspecting win-rates helps circumvent potential issues that may arise from applying BT regression to slices with fewer votes.

| | Front/Backend | Long Context | FiM | Non-Python |
|---|---|---|---|---|
| deepseek-coder | 0, -3 | +2, 0 | +1, 0 | 0, 0 |
| claude-3.5-sonnet | +4, 0 | 0, -1 | +2, 0 | +1, 0 |
| codestral | +1, 0 | +1, -1 | 0, 0 | 0, 0 |
| llama-3.1-405b | +1, -4 | +1, -1 | 0, 0 | 0, 0 |
| gemini-flash-002 | +1, -2 | 0, 0 | +1, -2 | 0, 0 |
| gemini-pro-002 | +1, 0 | +3, 0 | +2, 0 | 0, -1 |
| gpt-4o-2024-08-06 | +1, 0 | 0, -2 | 0, -2 | +1, 0 |
| llama-3.1-70b | +4, 0 | +1, 0 | +1, -2 | 0, 0 |
| qwen-2.5-coder-32b | 0, -2 | 0, -3 | 0, 0 | 0, -2 |
| gpt-4o-mini | +1, -3 | 0, 0 | 0, -1 | +1, 0 |
| % Total Changes: | 31.1 | 17.8 | 15.6 | 6.7 |

*Figure 7.* Significant win-rate changes as a result of different data partitions: frontend/backend versus algorithmic problems, long versus short contexts, FiM vs non-FiM, non-Python vs Python. We report the number of positive and negative changes (e.g., +1/-2 means that a model improved over 1 model and worsened against 2 models). In general, we observe the largest percentage of total changes as a result of differences in task, while the smallest effects as a result of differences in programming language.

our dataset. We stratify the data as follows: For tasks, we compare frontend/backend against algorithm design. For context length, we compare the top 20% against the bottom 20%. For FiM, we compare FiM against completion only. For programming languages, we compare all other programming languages against Python. We stratify these input features to highlight differences between the data distribution in Copilot Arena compared to static benchmarks (Table 1), where a positive win-rate indicates increased model performance on data that may be considered out of the distribution of typical static benchmarks. See Appendix E for full data on win-rates.

**Downstream task significantly affects win-rate, while programming languages have little effect.** Changing task type significantly affects relative model performance, with 28 significant win-rate changes (31.1% of all possible changes). This gap may indicate that certain models are overexposed to competition-style algorithmic coding problems. On the other hand, the effect of programming language on win-rates was remarkably small, resulting in only 6 (6.6%) significant changes, meaning that models that perform well on Python will likely perform well on another language. We hypothesize that this is because of the inherent similarities between programming languages, and learning one improves performance in another, aligning with trends reported in prior work (Peng et al., 2024). Context length and FiM have moderate effects to win-rate, which lead to 16 (17.8%) and 14 (15.6%) significant changes respectively.

**Smaller models tend to perform better on data similar to static benchmarks, while the performance of larger**

**models is mixed.** For example, Qwen-2.5 Coder performs noticeably worse on frontend/backend tasks (-2), longer contexts (-3), and non-Python settings (-2). We observe similar trends for the two other small models (Gemini Flash and GPT-4o mini) across multiple features. We hypothesize that overexposure may be particularly problematic for smaller models. On the other hand, performance amongst larger models is mixed. For example, Gemini-1.5 Pro performs noticeably better (+3) on long context which aligns with its goal of long context understanding (Team et al., 2024). However, Llama-3.1 405B underperforms on frontend/backend tasks (-4).

**Surprisingly, models explicitly trained for infilling do not experience large changes to win-rate.** Neither DeepSeek Coder, Codestral, nor Qwen-2.5 Coder sees any noticeable performance gains due to FiM. We run an experiment using DeepSeek Coder's Chat API with Snip-It (rather than FiM) and observe that relative model performance remains consistent (Table 6). These results suggest that Copilot Arena captures signals about code quality or usefulness rather than just formatting.

### 5.3. Insights and Takeaways

Based on our findings, we discuss two key insights and their implications for developing new models:

**Model performance is dictated by task, context, and code structure.** For example, while Claude 3.5 Sonnet excels at frontend and backend tasks, Gemini Pro and Deepseek outperform at longer contexts. With the exception of different programming languages, we observed that any other variation to the setting or use case affected model performance. Since there seems to be no model that is "one-size-fits-all," it is important for end users and developers of LLM-powered applications to choose their model depending on their use case. Alternatively, a routing approach based on input code context (Frick et al., 2025) is an interesting direction for future research.

**Human preference data is essential for effective code generation models.** The discrepancy between our leaderboard and static benchmarks indicates that current models are likely not trained on human preference data. Recent approaches have considered this, but still have significant gaps (e.g., Qwen-2.5 Coder trains on LLM-as-a-Judge preferences as a proxy for human preferences (Qwen et al., 2025)). We believe that training code models on human preference data is a promising future research direction. By releasing Copilot Arena data, we hope to help address the need for human preference data in real-world coding contexts.

## 6. Related Work

**Human Preferences for Evaluations.** A diverse set of human preferences—including binary preferences (Bai et al., 2022), fine-grain feedback (Wu et al., 2023; Kirk et al., 2024), and natural language (Scheurer et al., 2022)—are increasingly used for training and fine-tuning LLMs (Ouyang et al., 2022). Preferences are also important for human-centric evaluation, especially as LLMs are deployed in contexts that involve human interaction. Platforms like Chatbot Arena (Chiang et al., 2024) and Vision Arenas (Chou et al., 2024; Lu et al., 2024) provide a way for users to interact with LLMs and provide paired preference judgments. However, existing arenas lack integration into actual user environments to reflect the diverse data that may appear in a user's workflow. We study the use case of LLMs as coding assistants and introduce Copilot Arena to ground preference evaluations in a developer's working environment.

**Evaluations of LLM coding capabilities.** Static benchmarks, e.g., HumanEval (Chen et al., 2021) and MBPP (Austin et al., 2021), largely focusing on interview-style programming problems have been the most commonly used to evaluate coding capabilities (Lu et al., 2021; Nijkamp et al., 2023; Zhu et al., 2022; Wang et al., 2023; Liu et al., 2023; Jimenez et al., 2023; Khan et al., 2023; Yan et al., 2023; Cassano et al., 2023; Muennighoff et al., 2023; Dinh et al., 2023; Yang et al., 2024b), measured using `pass@k`. Recent benchmarks aim to create more realistic problems, which include multi-turn program evaluations (Nijkamp et al., 2023) and repository-level challenges (Jimenez et al., 2023; Jain et al., 2024b), and create live benchmarks that reduce contamination risks (Jain et al., 2024a; White et al., 2024). Our evaluation platform complements the existing suite of benchmarks by contextualizing model evaluations in an actual user's workflow as coding assistants, measuring a model's quality based on user preferences. Preference data retains signal when models output slightly incorrect, but still useful answers as opposed to a strict or all or nothing when evaluating using test cases.

A growing set of user studies aim to study human interactions with LLMs (Lee et al., 2023), particularly how programmers use LLM assistance for software development (Barke et al., 2023; Vaithilingam et al., 2022; Ross et al., 2023; Peng et al., 2023; Mozannar et al., 2024a; Murali et al., 2024; Chen et al., 2024). A notable work by Cui et al. (2024) conducted a field study on GitHub Copilot with many users. However, these studies generally face challenges of scale in terms of the number of users and the models considered, primarily relying on commercial tools like GitHub Copilot or ChatGPT. Mozannar et al. (2024b) conducted a study to evaluate six different LLMs of varying performance and Izadi et al. (2024) similarly conducted a study with three different LLMs, but the models evaluated

in both studies are no longer considered state-of-the-art. Our platform aims to address these challenges by building and deploying an actual coding assistant that allows for scalable and adaptable evaluation as new models emerge.

## 7. Discussion

**Limitations.** Although we have a diverse set of users and use cases, it is unclear to what extent our results encapsulate all real-world use cases. We run extensive pilot tests to ensure platform usability, but we recognize that certain aspects—specifically our pairwise completions and slower latency—do not perfectly mirror real-world platforms such as Github Copilot. Further, while we rank models based on user preferences, this should not be treated as the sole defining metric of model quality, but instead an informative one. In this work, we evaluate multiple LLMs with strong coding capabilities; however, we are unable to include Github Copilot because the model powering Github Copilot is not available via API. Additionally, since this is a live service, model performance directly affects user retention (and votes), rendering it difficult for us to evaluate weaker coding models. Finally, due to privacy considerations, we choose not to release all code contexts collected in the study without careful post-processing. We strive to make all data open through periodic releases.

**Future Work.** Our analyses of Copilot Arena data stress the need to create a diverse set of questions, including multiple written and programming languages, downstream applications, and code structures. Copilot Arena findings highlight the importance of conducting evaluations with real users, tasks, and environments. To extend this platform, future evaluations may also consider building on the Copilot Arena system in multiple ways: more nuanced forms of feedback in the programming setting, including measuring trajectories and code persistence metrics, and more forms of interaction, including inline prompt editing and chat dialogue within an IDE. We open-source Copilot Arena to facilitate these future extensions.

## 8. Conclusion

We introduce a platform, Copilot Arena, to evaluate LLMs in the wild using live human feedback for the use case of coding assistants. Copilot Arena is deployed and has collected over 11604 votes across 10 models; we will release a curated dataset to showcase the diversity of user preferences. We show that evaluating the coding capabilities of LLMs in Copilot Arena leads to rankings that differ from existing approaches which rely on static benchmarks or chat-based interactions, demonstrating how these differences could be attributed to the shift in distribution between Copilot Arena and prior evaluations. These different contexts also facil-

itate further understanding of how user preferences vary, highlighting the importance of evaluating new models with real users, tasks, and environments.

## Acknowledgments

This work was supported in part by the National Science Foundation grants IIS1705121, IIS1838017, IIS2046613, IIS2112471, and funding from Sony AI, Meta, Morgan Stanley, Amazon, Google, and Scribe. Any opinions, findings and conclusions or recommendations expressed in this material are those of the author(s) and do not necessarily reflect the views of any of these funding agencies.

## Impact Statement

This paper presents work whose goal is to advance the field of Machine Learning. There are many potential societal consequences of our work, none which we feel must be specifically highlighted here. However, we would like to highlight the ethical considerations of user privacy involved with storing and releasing user code data. We detail such considerations extensively in both Section 3 and Appendix B.

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

# A. Additional System Details

We describe further implementation details and considerations for each of the three key system components: user interface, model routing, and model prompting

## A.1. User Experience

We make several additional design decisions surrounding our user interface.

- We cache generated completions. If the user continues typing, we try to retrieve a matching pair of completions.

- We set a 0.5 second delay before automatically generating completions.

- If two completions are identical, then we return only one copy.

- If one model returns an empty string, then we only display the other, non-empty completion.

- We limit the number of lines each completion can generate (default of 20), but allow users to customize this limit.

- After the user votes, we reveal the model pair and their choice to the user. We also show the user a history of their votes.

- We limit the input file size to be 8,000 tokens, which covers nearly all user file lengths.

## A.2. Prompting Diverse Models

To enable the evaluation of a diverse set of models, we devise a prompting scheme to facilitate code completion functionality of models that were not necessarily trained to "fill-in-the-middle." Note that we focus on prompting-based approaches and do not fine-tune any LLMs, as many models we evaluate are commercial, closed models.

**Prompt templates.** The following are an overview of prompt templates from Gong et al. (2024) that we experimented with:[3]

1. *Prefix-Suffix-Middle (PSM)*. PSM presents the code context in the order of prefix and then suffix, using XML notation to demarcate prefix, suffix, and middle segments (e.g., `<PREFIX>` and `</PREFIX>`). The LLM is then asked to output the middle segment given the prompt.

2. *Suffix-Prefix-Middle (SPM)*. SPM is identical to PSM except that the suffix appears before the prefix, which may be more natural than having the suffix appear directly before the output as is the case with PSM.

3. *Mask*. Rather than using start and end tokens to denote the prefix and suffix, the Mask prompt uses a special "sentinel" token to indicate the masked (i.e. middle) code segment (Guo et al., 2024). The LLM is then requested to fill in the masked code segment.

4. *Instructed Prefix Feeding (IPF)*. IPF begins with the Mask prompt and then repeats the prefix as a "prefill" of the completion for the language model.[4] This is similar to IPF in Guo et al. (2024), except with instructions adjusted to better align with chat models. This approach allows non-FiM-trained models the ability to better tackle FiM tasks (Fried et al., 2023).

---

[3]Full prompt templates are provided in our code: `https://github.com/lmarena/copilot-arena`

[4]Nowadays, many completion APIs are deprecated; however, many chat APIs provide the ability to "pre-fill" tokens in the response which is similar to forcing the LLM to do a completion

## A.2.1. PSM EXAMPLE

```
Fill in code and output nothing else. Respect spacing, new lines, and
    indentation. Start with  and end with .
Be VERY mindful of indentation. Make sure it is correct.

Example 1:
<PREFIX>class Calculator {{
  add(number) {{
    this.result +=</PREFIX>
<SUFFIX>  subtract(number) {{
    this.result -= number;
    return this;
  }}
}}</SUFFIX>
 number;
    return this;
  }}

Example 2:
<PREFIX>from typing import List

def has_close_elements(numbers: List[float], threshold: float) -> bool:
    """ Check if in given list of numbers, are any two numbers closer to
        each other than
    given threshold.
    >>> has_close_elements([1.0, 2.0, 3.0], 0.5)
    False
    >>> has_close_elements([1.0, 2.8, 3.0, 4.0, 5.0, 2.0], 0.3)
    True
    """
    for idx, elem in enumerate(numbers):
        for idx2, elem2 in enu</PREFIX>
<SUFFIX> != idx2:
                distance = abs(elem - elem2)
                if distance < threshold:
                    return True

    return False</SUFFIX>
merate(numbers):
            if idx

Task:
<PREFIX>{prefix}</PREFIX>
<SUFFIX>{suffix}</SUFFIX>
```

### A.2.2. EVALUATION RESULTS

On these various templates, we evaluate 9 models on the HumanEval-infilling dataset (Bavarian et al., 2022), which consists of 1640 examples where random spans in a completed portion of code are masked out to simulate FiM behavior.

*Table 2.* pass@1 of code completions with different prompt templates (PSM, SPM, Mask). We observe that for all models and most prompt templates, our Snip-It method improves pass@1.

| Group | Model | psm | spm | mask | ipf | snip_psm | snip_spm | snip_mask | snip_ipf |
|---|---|---|---|---|---|---|---|---|---|
| **Open Code** | | | | | | | | | |
| | Deepseek-Coder-V2.5 | 0.551 | 0.519 | 0.414 | 0.229 | 0.585 | 0.584 | 0.597 | 0.614 |
| | Qwen-2.5-32B | 0.169 | 0.065 | 0.113 | 0.005 | 0.563 | 0.611 | 0.534 | 0.521 |
| **Open** | | | | | | | | | |
| | Llama-3.1-405B-Instruct-Turbo | 0.254 | 0.224 | 0.145 | 0.038 | 0.553 | 0.583 | 0.531 | 0.463 |
| | Llama-3.1-70B-Instruct-Turbo | 0.074 | 0.079 | 0.061 | 0.029 | 0.490 | 0.527 | 0.312 | 0.323 |
| **Commercial** | | | | | | | | | |
| | Gemini-1.5-Pro-002 | 0.620 | 0.599 | 0.562 | 0.338 | 0.561 | 0.659 | 0.259 | 0.491 |
| | GPT-4o | 0.607 | 0.477 | 0.505 | 0.033 | 0.620 | 0.670 | 0.609 | 0.524 |
| | Claude-3.5-Sonnet | 0.561 | 0.565 | 0.552 | 0.374 | 0.730 | 0.710 | 0.705 | 0.507 |
| | Gemini-1.5-Flash-002 | 0.434 | 0.376 | 0.277 | 0.286 | 0.409 | 0.403 | 0.301 | 0.394 |
| | GPT-4o-mini | 0.099 | 0.055 | 0.088 | 0.019 | 0.429 | 0.480 | 0.361 | 0.342 |

### A.2.3. ERRORS WITHOUT SNIP-IT

Below are two examples of errors without Snip-It from GPT-4o mini. Red indicates the incorrect code that the model filled in.

```python
from typing import List
def below_zero(operations: List[int]) -> bool:
"""  You're given a list of deposit and withdrawal operations on a bank account
    that starts with zero balance. Your task is to detect if at any point the
    balance of account fallls below zero, and at that point function should
    return True. Otherwise it should return False. >>> below_zero([1, 2, 3])
    False >>> below_zero([1, 2, -4, 5]) True  """
    balance = balance += op
        if balance < 0:
        balance += op
        if balance < 0:
            return True
        return False
```

```python
from typing import List
def parse_nested_parens(paren_string: str) -> List[int]: """ Input to this
    function is a string represented multiple groups for nested parentheses
    separated by spaces. For each of the group, output the deepest level of
    nesting of parentheses. E.g. (()()) has maximum two levels of nesting while
    ((())) has three. >>> parse_nested_parens('(()()) ((())) () ((())()())') [2,
    3, 1, 3] """
def parse_paren_group(s):
    depth += 1
            max_depth = max(max_depth, depth)
        if char == ')':
```

# B. User information

Below we provide a copy of the general instructions and privacy instructions for users.

*Table 3.* To conserve space, we refer models by *shortened name* in the main text but provide *full model name* below for completeness.

| Full Model Name | Shortened Name |
|---|---|
| deepseek-coder-fim | deepseek-coder |
| claude-3-5-sonnet-20240620 | claude-3.5-sonnet |
| codestral-2405 | codestral |
| llama-3.1-405b-instruct | llama-3.1-405b |
| gemini-1.5-flash-002 | gemini-flash-002 |
| gemini-1.5-pro-002 | gemini-pro-002 |
| gpt-4o-2024-08-06 | gpt-4o-2024-08-06 |
| llama-3.1-70b-instruct | llama-3.1-70b |
| qwen-2.5-coder-32b-instruct | qwen-2.5-coder-32b |
| gpt-4o-mini-2024-07-18 | gpt-4o-mini |

### B.1. General instructions

Step 1: Install the extension and restart Visual Studio Code after installation. If installed successfully, you will see Copilot Arena show up on the bottom right corner of your window and the check mark changes to a spinning circle when a completion is being generated, Note, if you are using any other completion provider (e.g. Github Copilot), you must disable them when using Copilot Arena.

Step 2: Copilot Arena currently supports two main feature: read autocomplete and in-line editing (beta) below to understand how to use each one. Since we show paired responses, the way you use them are slightly different than your standard AI coding tools!

Step 3: This step is optional. If applicable, you can change what data is saved by Copilot Arena by following the instructions in "Privacy Settings".

Step 4: Create a username by clicking the Copilot Arena icon on the sidebar; detailed instructions are also in "Create an account". Your username will be used for a future leaderboard to compare individual preferences.

### B.2. Privacy Instructions

**Privacy Settings.** Your privacy is important to us. Please read carefully to determine which settings are most appropriate for you. To generate completions, the code in your current file is sent to our servers and sent to various API providers. This cannot be changed.

**Data Collection.** By default, we collect your code for research purposes. You can opt-out of this. If you are working on code containing sensitive information, we recommend that you opt out of data collection. To opt-out of data collection, please change codePrivacySettings to Debug. We will only log your code for debugging. To disable logging entirely, please change codePrivacySettings to Private. Opting-out means any bugs you encounter will be non-reproducable on our end. You can find these settings by searching for Copilot Arena in your vscode settings or clicking the gear button of the Copilot Arena extension -> Extension Settings.

**Removing your data.** If you would like to have the option in the future for us to delete any of your data, you must create an account on Copilot Arena following instructions described in "Create an account." To remove your data, you can email any of the Copilot Arena maintainers with your username.

**Data Release.** Prior to releasing any collected code snippets to enable future research efforts, we will run a PII detector and remove any identified entities to further ensure no personal information is released.

## C. Data Analysis

**Natural Language Detection.** To detect natural languages, we used the lingua language detector (Stahl, 2024). We set the detector to all available languages (except for Latin due to false positives), and picked the language with the highest confidence that was greater than 0.7. Additionally, we filtered for languages that appeared at least 5 times. Results are

in Figure 8. For Table 1, since Chatbot Arena does not track natural languages, we ran the same detection algorithm for Chatbot Arena.

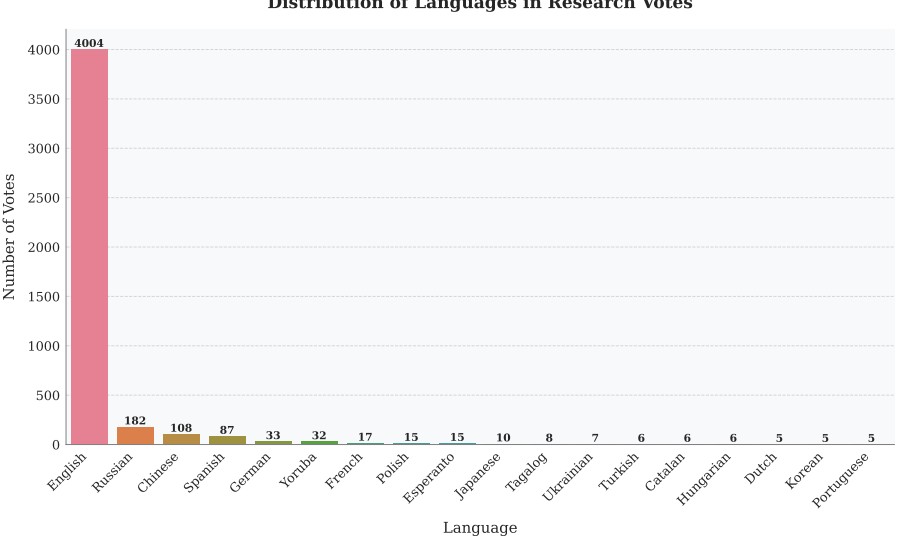

*Figure 8.* Natural languages in Copilot Arena

**Programming Language Detection.** We detect programming languages in Copilot Arena by using the file's extension type (Figure 9). For Table 1, since Chatbot Arena does not track programming languages, we checked for the language of codeblocks instead.

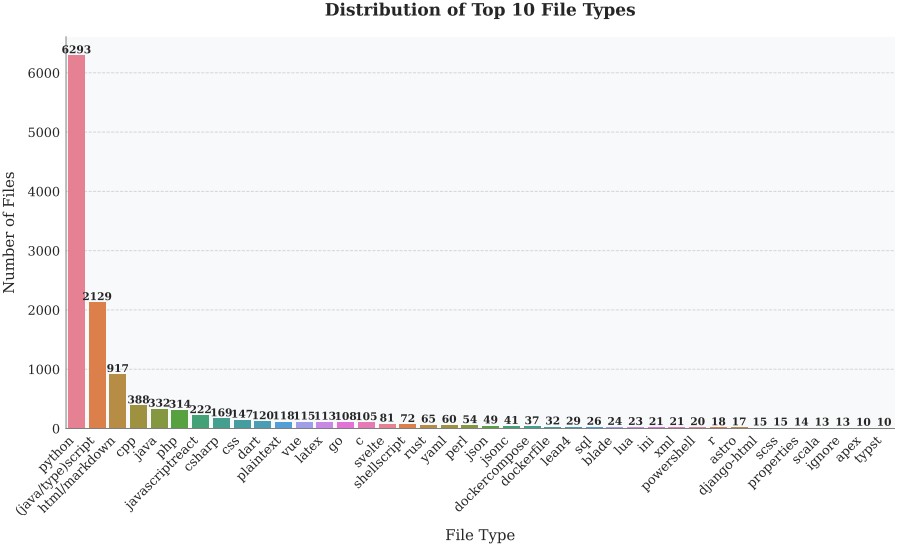

*Figure 9.* Programming languages in Copilot Arena. For image clarity, we only show programming languages that appear more than 10 times.

**Task detection.** Since Copilot Arena code contexts are fairly long, we employ a multi-step process to cluster code contexts, via LLM-as-a-judge (Zheng et al., 2023). We specifically use `GPT-4o-mini` due to its speed and price.

First, we summarize all code contexts into short one-sentence descriptions.

```
System Prompt
You are a helpful assistant that describes code files in a single, concise
sentence.  Focus on the main purpose and functionality of the code.  Keep
descriptions clear, technical, and under 100 characters.  Do not mention
file names or extensions in your description.

General Prompt
Describe this code in one sentence
```

Next, we prompt a model to cluster all one-sentence descriptions.

```
General Prompt

    You are a code organization expert.
    Analyze the provided code descriptions and:
    1. Identify 5-10 main functional clusters or themes
    2. Assign each description to the most appropriate cluster
    3. Provide a brief name and description for each cluster
    4. Format the response as valid JSON with the following structure:
    {
        "clusters": [
            {
                "name": "cluster_name",
                "description": "brief cluster description",
                "descriptions": ["description", "description2"]
            }
        ]
    }
```

Finally, we provide the full code context and ask the LLM to categorize the context given aforementioned clusters. Note that we sanity-checked clusters manually and removed redundant ones.

**System Prompt**
Please categorize the following code into one of these categories:

- User Interaction and Input Handling: Code that manages user inputs, prompts, and basic interaction with the system

- Frontend Development and UI Design: Code snippets focused on designing user interfaces and creating interactive components.

- Backend Development and APIs: Server-side logic, data management, and API integration for applications.

- Algorithm Design and Problem Solving: Code implementing algorithms to solve computational problems or optimize tasks.

- Data Processing and File Operations: Code that reads, writes, or processes data from files and other data sources.

- Game Development and Simulations: Code focused on creating games, simulations, and managing game dynamics.

- Artificial Intelligence and Machine Learning: Code related to AI and machine learning for training, inference, and application.

**General Prompt**
Only respond with the exact category name that best fits. No other text.
Here's the code:
```
[code content]
```

**Model Votes.** We ensured that our leaderboard has coverage across all models, where each model received at least 2,000 votes, as shown in Figure 10.

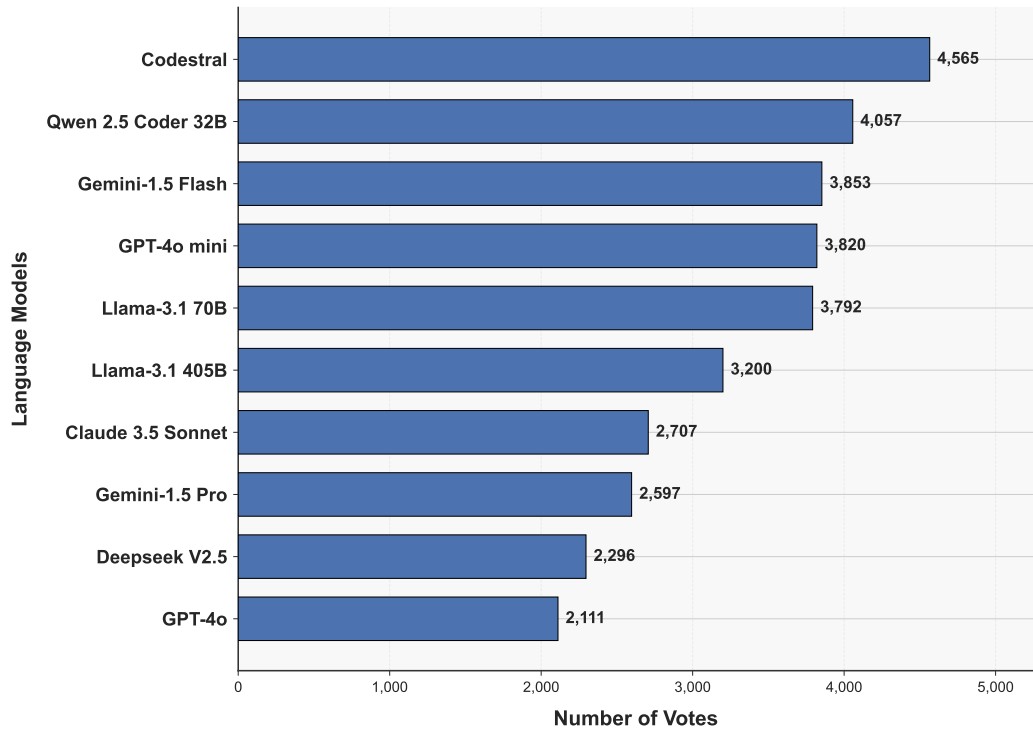

*Figure 10.* Number of votes for each language model.

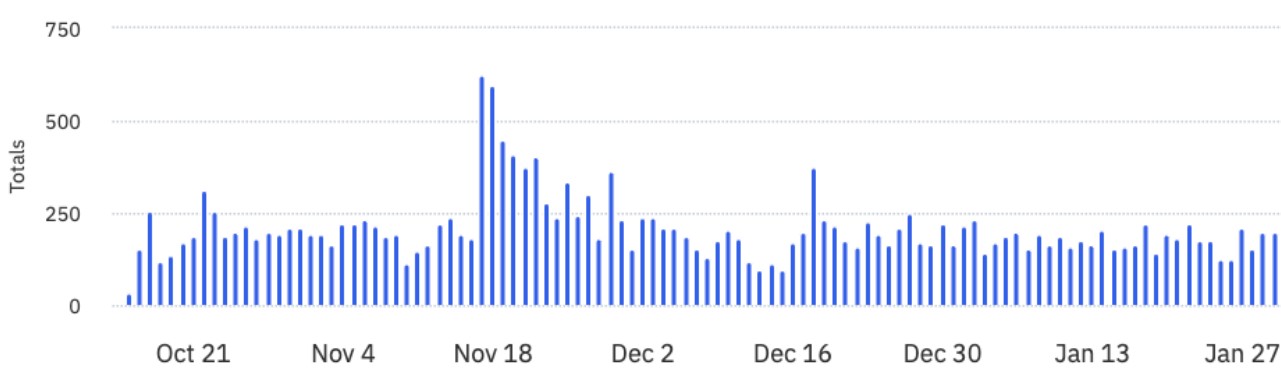

*Figure 11.* Number of votes over time.

**Code Structure Detection.** To detect the presence of FiM, we check if there exists a suffix. If there is only the prefix, we label it as "completion-only". To detect if there are comments, we check if any of the 5 previous lines start with common comment styles (e.g. #, //). We check for block comments in a similar fashion using docstring styles (e.g. """, /** */).

**Completion Bias.** Users selected the first completion 86% of the time, revealing a completion order bias. We investigated whether the bias was due to users instinctively pressing Tab for the first completion, as it requires a simpler keystroke than Shift-Tab. Analysis of decision times revealed that users spent a median of 6 seconds selecting the first completion, indicating this action was not automatic. However, users still took longer (9 seconds) to select the second completion, suggesting they deliberated more between the two options.

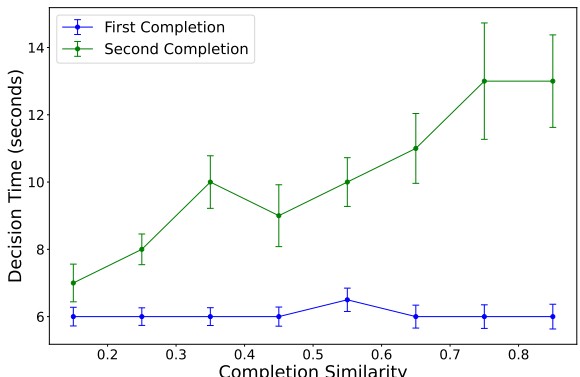

*Figure 12.* Completion similarity vs. decision time, grouped by selection of the first or second code completion.

We hypothesized that the extended deliberation resulted from users comparing completion differences. To validate this, we evaluated code similarity between completion pairs using the Levenshtein ratio. The dataset was refined by removing outliers (identical or very dissimilar completions) and excluding comments to minimize the impact of documentation differences. As shown in Figure 12, decision time increased with completion similarity for the second completion, indicating greater deliberation for highly similar completions. This trend was absent for the first completion, suggesting this extra deliberation did not occur for these cases.

**Cost.** We estimate that the total cost to collect our dataset is approximately $5k USD.

### C.1. Example Data

We provide examples of code contexts from each of the task categories. For readability, we select examples with shorter context lengths. Upon publication, we will also open-source more diverse examples (including those with significantly longer context lengths).

**Artificial Intelligence and Machine Learning**

```python
from main13 import knn, mlp
import pandas as pd

for pclass in [1, 2, 3]:
    for fare in range(10, 200, 10):
        my_df = pd.DataFrame({
                "Pclass": [pclass]*3,
                "Name": [24]*3,
                "Sex": [0]*3,
                "Age": [19]*3,
                "SibSp": [0]*3,
                "Parch": [0]*3,
                "Fare": [fare]*3,
                "Embarked": ["S", "Q", "C"]
            })
        my_df = pd.get_dummies(my_df, columns=["Embarked"], prefix="Embarked")
        my_df["Embarked_S"] = my_df["Embarked_S"].astype(int)
        my_df["Embarked_C"] = my_df["Embarked_C"].astype(int)
        my_df["Embarked_Q"] = my_df["Embarked_Q"].astype(int)

        predictions = {
            "knn": knn.predict(my_df),
            "mlp": mlp.predict(my_df)
        }
        ans_df = pd.DataFrame(index=[fare], columns=[1, 2, 3])
        ans_df.at[fare, pclass] = predictions
print(ans_df)
```

**Frontend Development and UI Design**

```html
<!DOCTYPE html>
    <html lang="en">
    <head>
      <meta charset="UTF-8">
      <meta name="viewport" content="width=device-width, initial-scale=1.0">
      <title>Document</title>
    </head>
    <body>
      <script>
        function getRandomNumber(min,max) {
          return Math.floor(Math.random()*)
        }
      </script>
    </body>
    </html>
```

**Algorithm Design and Problem Solving**

```java
import java.util.*;

public class hashmapImplementation {
    static class HashCode<K, V>{ //generics -> we can use any data type for
    key and value.
        private class Node{
            K key;
            V value;

            public Node(K key, V value){
                this.key = key;
                this.value = value;
            }
        }

        private int size; //n
        private LinkedList<Node> buckets[]; //N = buckets.length
        -> array of linkedlists

        @SuppressWarnings("unchecked")
        public HashCode() {
            this.size = 0;
            this.buckets = new LinkedList[4];
            for (int i = 0; i < buckets.length; i++) {
                buckets[i] = new LinkedList<>();
            }
        }

        public void put(K key, V value){

        }
    }

    public static void main(String args[]){
    }
}
```

**Data Processing and File Operations**

```python
import os
from pipeline.chain_function import *
path_name = "./pipeline_genereated_img/"

def upload_data(image_path):
    # image_path = os.listdir("pipeline_genereated_img")[0]
    full_path = path_name+image_path
    element = generate_img_summaries(full_path)

def upload_img_2_json():
# Write data to JSON file
json_file_path = "./img_json_stored/"+image_path
json_file_path = json_file_path.replace(".pdf",".json")
with open(json_file_path, 'w') as file:
    json.dump({"result": element}, file, indent=4)

print(f"Data successfully uploaded to {json_file_path}")
```

**User Interaction and Input Handling**

```python
print("Hello World")
namevar = input("Enter name ")

print("Welcome " + namevar)
#Write python code to download and run deepseek model locally in my windows
computer. I have python and pytorch installed in my computer.

#To download and run a DeepSeek model locally on your Windows computer,
you can follow these steps:
```

```
Game Development and Simulations
using System.Collections;
using System.Collections.Generic;
using UnityEngine;

public class Player : MonoBehaviour
{
    // Start is called before the first frame update
    public float speed = 1f;

    void Start()
    {

        transform.position = new Vector3(0,0,0);
    }

    // Update is called once per frame
    void Update()
    {
        transform.Translate(Vector3.left * speed * Time.deltaTime );
    }
}
```

```
Backend Development and APIs
async def discover_device(emp_user_no, access_token, repositories):
    print(f"dicsover_device")

async def check_device_health(request_type, payload, mesage_id, emp_user_no,
repositories)
    print(f"check_device_health")

async def
```

## D. Details on Model Ranking

**Computing BT Coefficients.** We estimate $\hat{\beta}$ by running a logistic regression:

$$\hat{\beta} = \arg \min_{\beta \in \mathbb{R}^M} \frac{1}{n} \sum_{i=1}^{n} \text{CE}(\sigma(X_i^\top \beta), Y_i) \tag{3}$$

where CE represents the cross-entropy loss and $\sigma$ is the sigmoid function. We use the sklearn package with l2 penalty and no intercept term. We bootstrap the ranking calculation by sampling with replacement for 100 rounds to compute the 95% confidence interval. In our leaderboard, we use codestral as an anchor model.

**BT Ablations.** Since confounding variables (e.g., length of the response or other stylistic formatting (Singhal et al., 2023)) may influence preference judgments, we also control for these variables in the BT model. Given a set of style features, which include model latency and completion length, we add a style vector to the BT model $\vec{Z}$ where= $Z_i \in \mathbb{R}^S$ is a vector of $S$ style features comprising the normalized difference between the feature values of both model responses. The extended

BT model includes the style coefficients $\gamma \in \mathbb{R}^S$ and can be written as:

$$\hat{\beta}, \hat{\gamma} = \arg \min_{\beta \in \mathbb{R}^M, \gamma \in \mathbb{R}^S} \frac{1}{n} \sum_{i=1}^{n} \mathrm{CE}(\sigma(X_i^\top \beta + Z_i^\top \gamma), Y_i)$$

where CE represents the cross-entropy loss and $\sigma$ is the sigmoid function. The resulting $\hat{\beta}$ represents model strengths adjusted for style effects, while $\hat{\gamma}$ quantifies the influence of style on user preferences. $\hat{\beta}$ values are used to create the ordered ranking of models on the leaderboard.

Since we observe a completion bias (described in Appendix C), we perform an additional experiment to verify that we collected sufficient votes across all pairs. To do so, we upsampled and downsampled model pairs that had fewer and more comparisons, respectively, so that the difference between the most and least voted pair is within 10%. We find that the rankings remain identical to the ranking obtained using the full set of votes. This provides additional confidence in our reported leaderboard.

When comparing the original leaderboard (Table 4) and the style-controlled version (Table 5), we see minimal changes to the overall "tiers" described in the main text. While we observe some changes in the middle tier (e.g., Llama-3.1-405b, Gemini-1.5-Flash, and Gemini-1.5-Pro swap places as well as GPT-4o and Llama-3.1-70b), we do not observe significant changes *between* tiers.

Table 4. $\beta_i$ values for each model bootstraped over 100 samples: their lower, rating, and upper bounds.

| Model | Lower bound | $\beta$ estimate | Upper bound |
|---|---|---|---|
| deepseek-coder-fim | 0.04 | 0.07 | 0.10 |
| claude-3-5-sonnet-20240620 | 0.02 | 0.06 | 0.09 |
| codestral-2405 | -0.02 | 0.00 | 0.02 |
| llama-3.1-405b-instruct | -0.07 | -0.04 | -0.01 |
| gemini-1.5-flash-002 | -0.06 | -0.04 | -0.01 |
| gemini-1.5-pro-002 | -0.08 | -0.05 | -0.02 |
| gpt-4o-2024-08-06 | -0.09 | -0.06 | -0.03 |
| llama-3.1-70b-instruct | -0.10 | -0.07 | -0.04 |
| qwen-2.5-coder-32b-instruct | -0.16 | -0.13 | -0.10 |
| gpt-4o-mini-2024-07-18 | -0.19 | -0.15 | -0.12 |

Table 5. $\beta_i$ and $\gamma_i$ values for each model bootstraped over 100 samples: their lower, rating, and upper bounds.

| Model | Lower | Rating | Upper |
|---|---|---|---|
| deepseek-coder-fim | 0.05 | 0.08 | 0.11 |
| claude-3-5-sonnet-20240620 | 0.03 | 0.06 | 0.09 |
| codestral-2405 | -0.02 | -0.00 | 0.03 |
| gemini-1.5-flash-002 | -0.06 | -0.03 | -0.01 |
| llama-3.1-405b-instruct | -0.07 | -0.04 | -0.00 |
| gemini-1.5-pro-002 | -0.09 | -0.05 | -0.01 |
| llama-3.1-70b-instruct | -0.09 | -0.06 | -0.03 |
| gpt-4o-2024-08-06 | -0.09 | -0.07 | -0.04 |
| qwen-2.5-coder-32b-instruct | -0.16 | -0.13 | -0.10 |
| gpt-4o-mini-2024-07-18 | -0.18 | -0.15 | -0.12 |

| Model | Lower bound | $\gamma$ estimate | Upper bound |
|---|---|---|---|
| Model latency | -0.33 | -0.17 | 0.00 |
| Response length | 0.11 | 0.21 | 0.32 |

# E. Additional Results

We provide additional details about win-rate analysis in Figure 13, 14, 15, and 16. We also showcase an experiment testing our prompting approach in Table 6

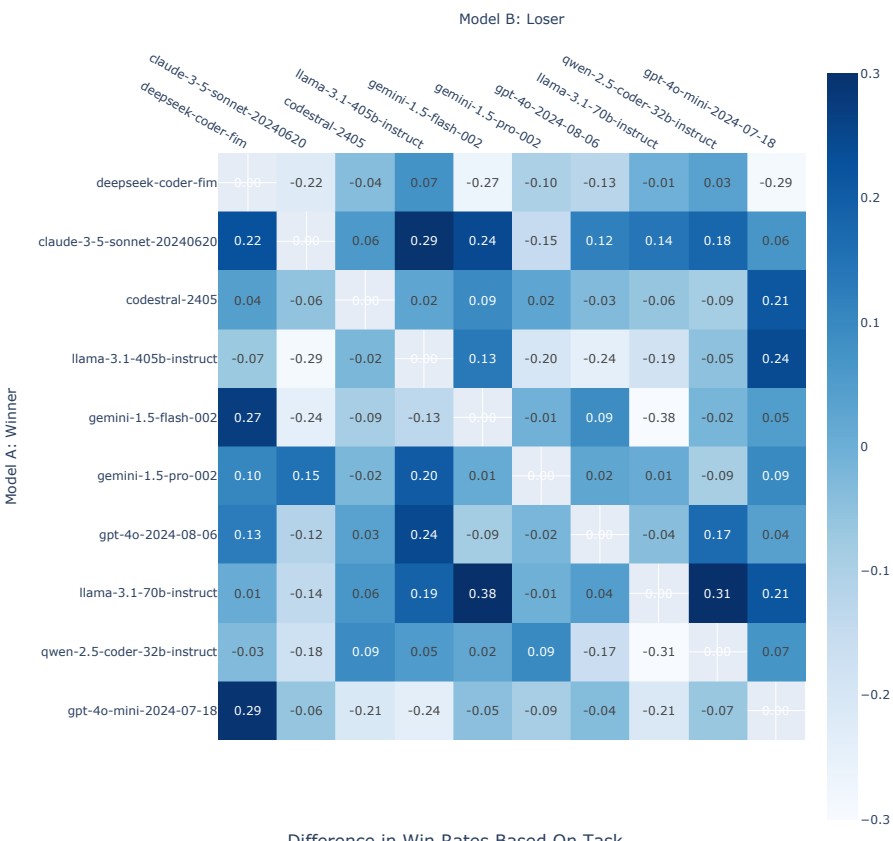

*Figure 13.* Win-rate difference based on Task: frontend/backend versus algorithmic design problems.

*Table 6.* A controlled experiment with a fixed model, DeepSeek Coder, where we vary whether we use the model's FiM capability or we use Snip-It to post-process the model as we would with other models that do not have native FiM capability. While we were not able to obtain a significant number of votes before deepseek-coder was deprecated, we still observe that $\beta$ estimates are comparable between the two variants. This shows that our Snip-It approach can roughly recover FiM capabilities.

| Model | Lower bound | $\beta$ estimate | Upper bound |
|---|---|---|---|
| deepseek-coder-fim | 0.04 | 0.07 | 0.10 |
| deepseek-coder | -0.04 | 0.06 | 0.15 |

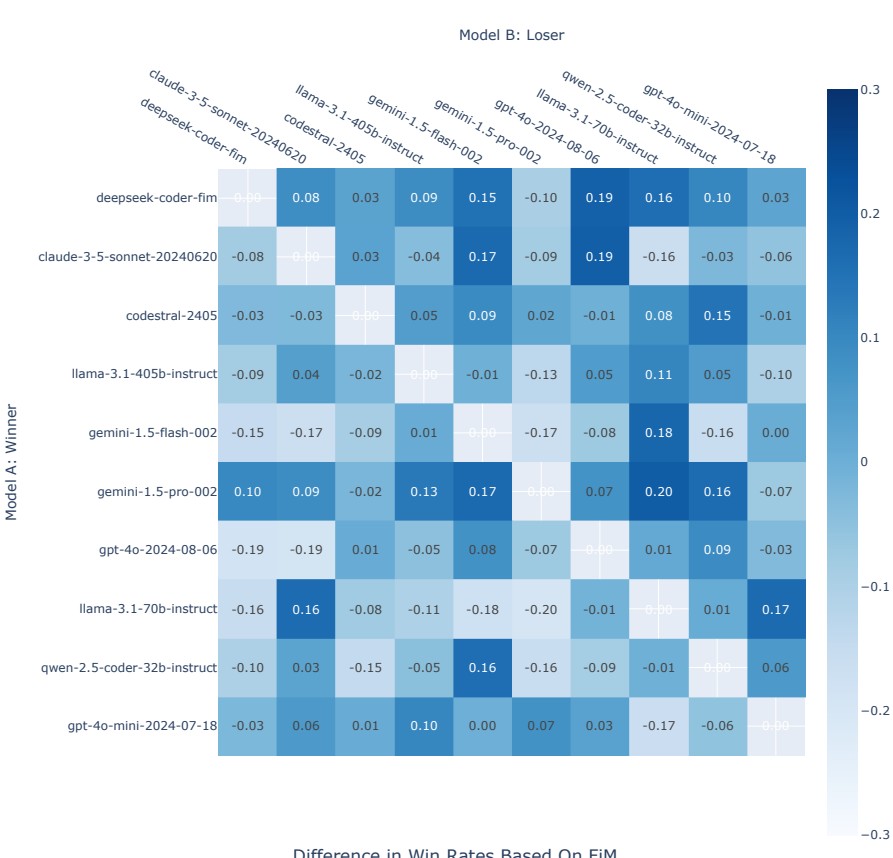

*Figure 14.* Win-rate difference based on FiM: whether the task is FiM or not.

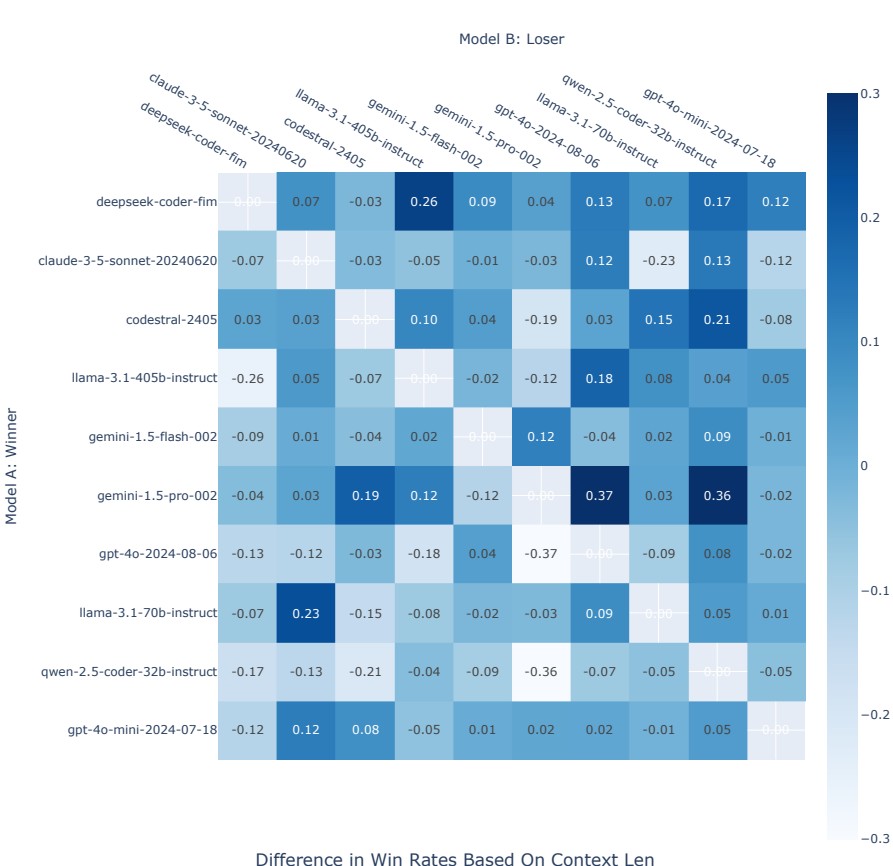

*Figure 15.* Win-rate difference based on context length: context length in top versus bottom 20 percentile.

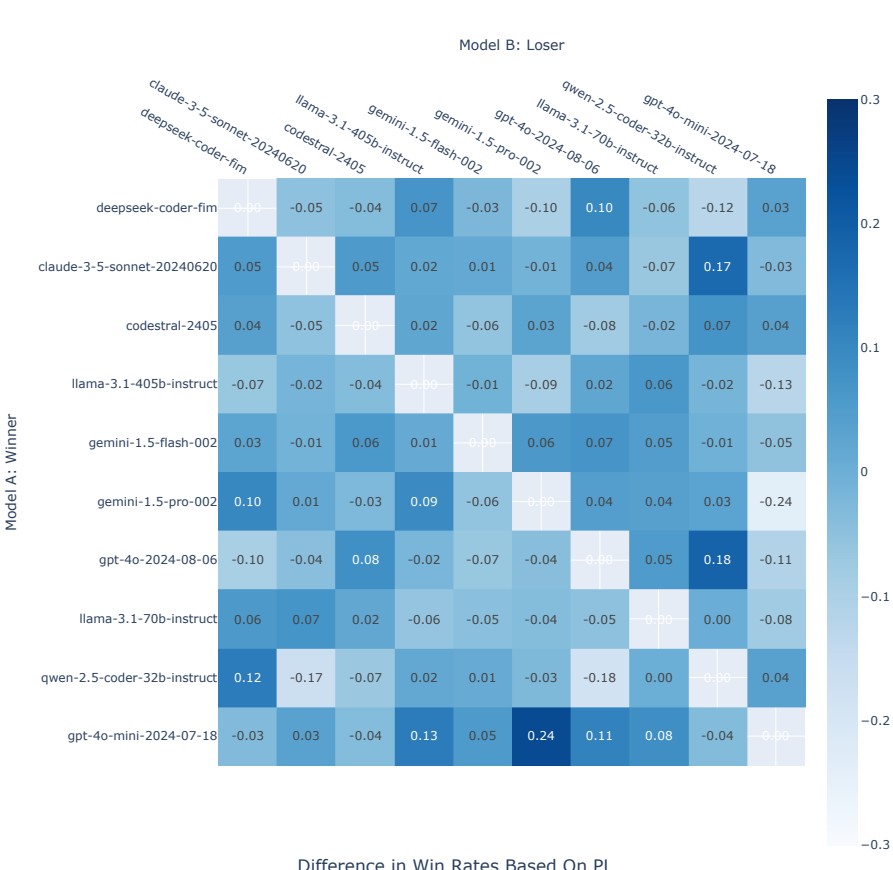

*Figure 16.* Win-rate difference based on programming language (PL): Non-python code versus Python code.

