# OpenReview forum: "Copilot Arena: A Platform for Code LLM Evaluation in the Wild"
_ICML.cc/2025/Conference — ICML 2025 poster_

### Official Review · Reviewer_mirs · 2025-03-12

**Overall Recommendation:** 3

**Summary:**

This paper presents EvalX, a platform for evaluating coding LLMs in real-world environments. Integrated into developers' IDEs, it collects user preferences on code completions. Unlike static benchmarks, EvalX provides real coding tasks and optimizes latency. Findings show model rankings differ from traditional evaluations, highlighting real-world coding insights.

**Claims And Evidence:**

Yes.

**Essential References Not Discussed:**

No.

**Experimental Designs Or Analyses:**

Yes. By collecting user preferences, the author evaluated the models.

**Methods And Evaluation Criteria:**

Yes.

**Other Comments Or Suggestions:**

No.

**Other Strengths And Weaknesses:**

**Strengths**
1. This work proposes a novel method for llm code evaluation, which collects user preferences for models through a vscode extension.
2. Compared with static code evaluation benchmarks, this method can better reflect users' preferences for llm in real scenarios.

**Weaknesses**
1. The evaluation metric of the model is relatively simple and can only reflect user preferences.
2. Although the benchmark is multilingual, the uneven use of programming languages ​​by users may result in inaccurate benchmarks for less common languages.

**Questions For Authors:**

Can this method include evaluations of some open-source models? Doing so seems to negatively impact the user experience. How do you think you should balance user experience with incorporating more models into the evaluation?

**Relation To Broader Scientific Literature:**

This work is related to the evaluation of code generation for large models.

**Theoretical Claims:**

N/A

---

> ### Author Rebuttal · Authors · 2025-04-01
>
> Thanks for your helpful comments. We address your comments below:
>
> *[Weakness 1: The evaluation metric of the model is relatively simple and can only reflect user preferences.]*
> - There is an extensive list of existing literature that follows this paradigm (see the first paragraph of the related work section). Despite its simplicity, it has proven to be an effective way to have humans-in-the-loop to evaluate models
> - However, the data we collect enables other forms of evaluation. For example, by utilizing the snapshots from a user’s code trajectory, we can analyze the long-term impact of each code completion. This is a direction we are actively pursuing as future work.
>
> *[Weakness 2: Although the benchmark is multilingual, the uneven use of programming languages ​​by users may result in inaccurate benchmarks for less common languages.]*
> - We agree with your concern but note that we report model performance aggregating across languages, rather than performance on an individual language.
> - Your comment prompted us to do further analysis into the distribution across languages. We found that even accounting for examples with over 50+ samples, we still have 23 programming languages. This is significant compared to previous static benchmarks (e.g., those mentioned in Table 1). Please see the following table for a more thorough distribution of our programming languages:
> - We propose to add this table and modify the writing. For example, in our data analysis (Section 5.1), we can explicitly discuss how the data is not evenly distributed over all 103 languages, but there is a core set of languages for which there are a substantial number of votes. We believe these changes will help reduce any misinterpretations of our results on multi-lingual programming languages.
>
> | Vote Count | # Programming Languages |
> |---------------|----------------------|
> | 5 | 65 |
> | 10 | 45 |
> | 25 | 31 |
> | 50 | 23 |
> | 100 | 17 |
>
>
> *[Question 1: Can this method include evaluations of some open-source models? How do you think you should balance user experience with incorporating more models into the evaluation?]*
> - Yes, EvalX can include open-source models! In fact, we reported results in our submission on 4 open-source models from multiple organizations (e.g., Llama 70b [1], Llama 405b [2], Qwen32b Coder[3], Codestral [4]). Since EvalX is an ongoing data collection effort, we continue to add open-source models to our platform.
> - However, you do raise a great point that we will clarify in our revision. We don’t anticipate being able to evaluate all models using EvalX because we do not want to impact user experience negatively. One way to operationalize this is to select models that perform well on existing benchmarks, indicating they will be usable in practical settings. This is what we did when selecting models for this work.
>
> [1] https://huggingface.co/meta-llama/Llama-3.1-70B.
> [2] https://huggingface.co/meta-llama/Llama-3.1-405B
> [3] https://huggingface.co/Qwen/Qwen2.5-Coder-32B-Instruct
> [4] https://huggingface.co/mistralai/Codestral-22B-v0.1

---

### Official Review · Reviewer_1UGk · 2025-03-13

**Overall Recommendation:** 3

**Summary:**

This paper discusses EvalX, a system deployed in-the-wild to gather human preferences regarding code. It constructs a leaderboard based on user preferences and identifies notable differences compared to existing static benchmarks and human preference leaderboards. By analyzing EvalX’s diverse and unique data distribution, this study derives new insights into user preferences for code.

## Update after rebuttal
The contribution is ok for me. This paper is acceptable.

**Claims And Evidence:**

Well supported.

**Essential References Not Discussed:**

No

**Experimental Designs Or Analyses:**

The soundness is pretty good.

**Methods And Evaluation Criteria:**

Yes. It's reasonable.

**Other Comments Or Suggestions:**

I recommend rephrasing Section 2.3, as the organization of the entire subsection is difficult to follow. Additionally, sentences like "we use offline datasets to improve chat models’ infilling capabilities" are not easy to comprehend. Could you clarify whether the offline datasets are used for tuning the models?

The observation that "smaller models seem to outperform in other static benchmarks compared to our leaderboard" might be attributed to the FiM task involved in the evaluation. I suggest further clarification of the underlying causes.

**Other Strengths And Weaknesses:**

I have some concerns regarding the scope of this paper. While the deliverables are well-suited for the community, the main methodology appears to offer limited contribution to this ML community. Despite these concerns, I am inclined to accept the paper.

**Questions For Authors:**

How do you determine whether a completion is a FiM task? Although there might be a suffix, the completion may not be related to it. Classifying it as a FiM task affects the evaluation of model performance.

How do you categorize the domain of the completion, such as frontend or backend?

Is it particularly challenging to further scale the collected data in terms of increasing the data size for a specific language?

**Relation To Broader Scientific Literature:**

Provide a dataset in the real world.

**Theoretical Claims:**

NA

---

> ### Author Rebuttal · Authors · 2025-04-01
>
> Thank you for the helpful suggestions. We address the comments below:
>
> *[Weakness 1: While the deliverables are well-suited for the community, the main methodology appears to offer limited contribution to this ML community. Despite these concerns, I am inclined to accept the paper.]*
>
> We thank the reviewer for advocating for accepting our work. We believe there are multiple reasons why the EvalX is a timely and important contribution to the ML community. Since the final version allows an additional page, we propose to add a Section 5.3 to explicitly summarize the insights and takeaways for researchers building new coding assistants:
>
> - **Different models excel in different settings.** Claude 3.5 Sonnet performs better at frontend/backend tasks, while Gemini and Deepseek excel at longer contexts. In contrast, changing models for different programming languages seems unnecessary. More generally, a routing approach based on input code context is an interesting direction for future research.
> - **Models should be trained and evaluated on varying code structures.** We observed a variety of tasks and code structures in our platform (e.g., FiM, docstrings, and inline comments). Future benchmarks and training schemes should explicitly account for variations in code structures.
> - **Models should be trained on human preference for code.** Given the clear gap between our leaderboard and static benchmarks, models should be trained on human preferences (of which our dataset is one option). Recent approaches have considered this, but still have significant gaps (e.g., Qwen-2.5 Coder trains on LLM-as-a-Judge preferences as a proxy for human preferences). By releasing EvalX data, our work can help to address the need for human preference data in real-world coding contexts.
>
> *[Suggestion 1: I recommend rephrasing Section 2.3….clarify whether the offline datasets are used for tuning the models?]*
> - As discussed in L166 (right), we do not fine-tune models, but rather only use the dataset for evaluation and tuning our prompts (L195, left).
> - However, we agree that Section 2.3 can be further clarified! To this end, we will also significantly expand on existing details in Appendix A and release all corresponding code for these experiments.
> - Altogether, with changes to Section 2.3 writing, additional details in Appendix A, and full release of code, we believe this will reduce any confusion on our prompting methodology.
>
> *[Suggestion 2: explain FiM task relation to "smaller models seem to outperform in other static benchmarks compared to our leaderboard".]*
> - In Section 5.2, our analysis indicates that FiM may not be the main contributor to model performance. While direct comparisons between models with and without FiM training would be ideal, most providers don't disclose their training paradigms. Our ablation using the Deepseek API (Appendix E, Table 6) demonstrates that FiM as an input format doesn't significantly impact performance. We'll clarify this nuance in our revised text.
>
> *[Question 1: How do you determine whether a completion is a FiM task?]*
> - To answer this, we conducted further analysis on whether the suffix is related to the completion. We broadly categorized our suffixes as either 1) inline (in which case it is clearly related) or 2) on a new line. Suffixes on the next line could be in scope (i.e., in the same function or loop as the last line of the prefix) or out of scope.
> - As shown in the table below, the vast majority (~80%) of our suffixes are related to the completion. We will include this analysis in the Appendix.
>
> | Formatting | Scope         | Percentage |
> |------------|---------------|------------|
> | Newline    | Out of Scope  | 20.4%      |
> | Newline    | In Scope      | 22.2%      |
> | Inline     |  In Scope        | 57.3%      |
>
> *[Question 2: How do you categorize the domain of the completion, such as frontend or backend?]*
>
> We follow a two-step process detailed in the Appendix (starting from L880), which we summarize here:
> - First, we ask a model (e.g., GPT-4o-mini) to summarize all code contexts into short one-sentence descriptions.
> - Next, we prompt a model (e.g., GPT-4o) to cluster all one-sentence descriptions.
> - Finally, we provide the full code context and ask the model to categorize the context given the aforementioned clusters.
> We additionally note that two authors of the work verified that the categorizations were sensical before scaling.
>
> *[Question 3: Is it particularly challenging to further scale the collected data ... for a specific language?]*
> - Since we are collecting data in the wild, we cannot directly control what languages we collect. However, we strive to grow the user base of EvalX to reach broader audiences, which will lead to more data in additional programming and natural languages.

---

> > ### Comment · Reviewer_1UGk · 2025-04-03
> >
> > I find the response satisfactory. Please incorporate the details into the accepted version. I will not change my rating.

---

### Official Review · Reviewer_tJ9Z · 2025-03-18

**Overall Recommendation:** 3

**Summary:**

The paper presents EvalX, a platform for evaluating coding capabilities of large language models (LLMs) in real-world settings. Unlike existing evaluations that rely on synthetic benchmarks or chat-based interactions, EvalX integrates directly into developers' VSCode environments to collect authentic user preferences on code completion pairs.

Key contributions include: (1) a novel interface for comparing model outputs directly in the IDE, (2) a sampling strategy to reduce latency, (3) a prompting scheme to enable FiM code completion functionality, and (4) insights into user preferences across different programming contexts. The authors collected over 11k pairwise judgments across 10 models and found differences between their leaderboard and existing static benchmarks.

**Claims And Evidence:**

Most claims are reasonably supported, with several issues:
- The position bias in the interface design is concerning - with 86% of users selecting the first completion (requiring just Tab vs. Shift+Tab). While the authors acknowledge this bias and analyze decision times (median 6s for first completion, 9s for second), they don't sufficiently address how this fundamental asymmetry might invalidate preference data. Lots of the preference data may reflect convenience rather than quality assessment, which may lower the data quality.

- While the authors claim supporting 103 programming languages, the actual distribution is skewed. Python alone accounts for 6000+ samples, while many languages have minimal representation.

**Essential References Not Discussed:**

I think this paper might also be related to some repo-level completion evaluation works, e.g.,:
-  CrossCodeEval: A Diverse and Multilingual Benchmark for Cross-File Code Completion (NeurIPS 2023)
- RepoBench: Benchmarking Repository-Level Code Auto-Completion Systems (ICLR 2024)
- REPOEXEC: Evaluate Code Generation with a Repository-Level Executable Benchmark (seems still under review)
etc..

But it seems hard to directly compare but may worth mentioning

**Experimental Designs Or Analyses:**

The user study design shows impressive scale and ecological validity.

**Methods And Evaluation Criteria:**

The methods employed are novel and address a real need in the field. The in-the-wild evaluation approach is valuable, and the Bradley-Terry model for ranking is appropriate.

- While optimizing for lower latency improves user experience, the sampling strategy potentially undersamples specific model pairs, is there any way to quantify the potential impact on the reliability of specific model pair comparisons (e.g., will the lack of certain pairs)?

**Other Comments Or Suggestions:**

N/A

**Other Strengths And Weaknesses:**

### Strengths:

- The "Snip-It" method enabling non-FiM models to perform FiM tasks is an innovative contribution beyond the evaluation platform itself.
- The analysis of different factors influencing preferences (Figure 7) provides valuable insights for future model development.
- Extensive ablation studies.
- Valuable dataset

### Weakness:
- see above

**Questions For Authors:**

See above.

**Relation To Broader Scientific Literature:**

The work makes a significant contribution by bridging static benchmarks and human preference evaluation platforms, providing super valuable human preference data (manual labelled). It extends evaluation methodologies from chat-based platforms like Chatbot Arena.

**Theoretical Claims:**

The paper appropriately focuses on empirical findings rather than theoretical claims. The mathematical formulation of the sampling strategy (equations 1 and 2) is clearly presented, and the Bradley-Terry model application is sound.

---

> ### Author Rebuttal · Authors · 2025-04-01
>
> Thank you for the helpful suggestions. We address your comments below:
>
> *\[Claim \+ Evidence 1: The position bias in the interface design is concerning. The preference data may reflect convenience rather than quality assessment, which may lower the data quality.\]*
>
> * Since we randomize the ordering of the two generated responses (L149), the position bias affects **all** models equally. This means that the overall **quality** of the assessment is not impacted. However, although the **quality** of the assessment is not impacted, the **efficiency** of our platform is impacted. In short, since there’s a significant position bias, we need more votes to shrink the confidence intervals enough to draw reliable conclusions.  Recall, we use logistic regression to estimate Bradley-Terry coefficients and bootstrap the samples to build confidence intervals (which allow us to be able to tell whether models are statistically significantly different). Overall, despite the efficiency reduction from positional bias, EvalX collects a sufficient number of votes to yield tight confidence intervals on model comparisons.
>
> *\[Claim \+ Evidence 2:  While the authors claim supporting 103 programming languages, the actual distribution is skewed.\]*
>
> * We agree with your concern but note that we report model performance *aggregating* across languages, rather than performance on an individual language.
> * Your comment prompted us to do further analysis of the distribution across languages. We found that even accounting for examples with over 50+ votes, we still have 23 programming languages. This is significant compared to previous static benchmarks (e.g., those mentioned in Table 1). Please see the following table for a more thorough distribution of our programming languages:
> * We propose to add this table and modify the writing. For example, in our data analysis (Section 5.1), we can explicitly discuss how the data is not evenly distributed over all 103 languages, but there is a core set of languages for which there are a substantial number of votes. We believe these changes will help reduce any misinterpretations of our results on multi-lingual programming and natural languages.
>
> | Vote Count | \# Programming Languages |
> |---------------|----------------------|
> | 5 | 65 |
> | 10 | 45 |
> | 25 | 31 |
> | 50 | 23 |
> | 100 | 17 |
>
> *\[Method 1: Is there any way to quantify the potential impact on the reliability of specific model pair comparisons?\]*
>
> * In the analysis conducted in the original submission, we took a few steps to mitigate this impact.
>   * First, when computing Bradley-Terry coefficients, we use an L2 regularization term to prevent overfitting and bias towards models that have received more votes.
>   * Second, we conduct a statistical bootstrap of the leaderboard so we can obtain confidence intervals around the estimated coefficients, which improves the reliability of results.
>   * Finally, we ensure that we have reasonable coverage across all model pairs (e.g., \>150 votes per pair), so votes are not too sparse across some pairs.
> * Inspired by the author's suggestion, we did additional simulations using our data. We upsampled model pairs that had fewer comparisons and downsampled model pairs that had more comparisons so that the difference between the most and least voted pair is within 10%. We find that the rankings remain **identical to those** we report in the original submission. This provides additional confidence that we collected sufficient votes across all pairs.
> * Overall, we appreciate this suggestion and will highlight this more prominently as a point of consideration for future evaluation platforms.
>
> Finally, thank you for the helpful pointers to additional evaluation benchmarks\! We agree it would be challenging to include in our comparisons, but we will definitely mention them in the related work and opportunities for extending EvalX.

---

### Official Review · Reviewer_tbZQ · 2025-03-20

**Overall Recommendation:** 3

**Summary:**

Authors introduce EvalX a platform to compare the effectiveness of different LLMs for the use case of coding assistants. Their deployed platform has already collected over 11000 responses on comparisons between 10 different models. The model ranking presented from these results gives new insights on user preferences under different tasks that differ from the previous benchmarks.

**Claims And Evidence:**

Authors advocate the need for a coding specialized benchmark beyond existing works like Chatbot Arena. While I agree with this claim, I am afraid their exclusive focus on code completion with the FIM/L2R setup fails to capture the entire spectrum of AI tools used in the IDE. Most leading developer AI assistants (Github Copilot, Cursor) today provide the LLM interface in multiple formats beyond standard L2R or FIM completion, particularly the chat functionality (where for instance users can understand a repo by including it as a context). I'm not certain of the relative usage of such features (code completion vs chat), and a discussion on this aspect can immensely benefit this paper. But I find the sole emphasis on code completion to be a limitation of this work, as this does not throw light on the effectiveness of a candidate model in the chat setting. In my assessment, this work has immense value for the research community, but it is crucial for the paper to accurately convey what this benchmark captures (code completion capabilities), and what it does not (chat style functionalities). This could perhaps be accomplished by specifically referring to the capabilities being assessed rather than claiming wholistic real-world evaluation of Code LLMs. This is only briefly expressed towards the end of the paper (Section 7) currently.

Secondly, authors propose that LLM evaluation can be entirely based on human preference evaluation, but this may not necessarily be true. Say model A gives a more readable but less performant piece of code than model B, and the user selects model A's output under EvalX - that does not necessarily reflect the superiority of model A over B? User preference may be biased towards readability while ignoring aspects like efficiency (runtime and space), maintainability, and security vulnerability of generated code. While static benchmarks do not represent real world use cases well enough, they can be adapted to capture these attributes of LLM generated code, and they offer significant value in that regard, which is difficult to record via human preferences. EvalX's fine-grained evaluation (user feedback on specific completions) also ignores aspects like the long term impact of LLMs on developer productivity (e.g. by what factor is the time required to complete a project reduced with a specific model behind a coding assistant).

**Essential References Not Discussed:**

No.

**Experimental Designs Or Analyses:**

- Given that users are less likely to vote when experiencing high latency, authors have optimized model pair sampling to reduce the latency observed with some regularization to stay close to uniform.
    - The data is skewed towards faster models this way - does that not impact the quality of the resulting dataset? And the observations you can draw from it?

Related questions:
   - How to quantify the impact of this model sampling scheme - What'd be the difference in response/ranking rate with your model sampling vs uniform sampling?
    - How would tweaking $\tau$ impact coverage and user responsiveness?

**Methods And Evaluation Criteria:**

In my assessment the proposed methods and evaluation make sense for the problem at hand.

**Other Comments Or Suggestions:**

- Other benchmarks could be described with a little more detail to help readers understand how EvalX fits into the broader context of LLM evaluation

- Lack of insights/takeaways for model training or prompting
    - Would you advocate specialized (perhaps smaller models) over general (large models) for developers?
    - I believe the paper could immensely benefit from insights or recommendations for users of coding assistants and model builders (of LLMs powering AI assistants)

- Minor:
    - Chatbot Arena citation in Section 1
    - Takeaways not clear in Section 1

**Other Strengths And Weaknesses:**

Strengths
- Very promising and convincing approach on comparing and evaluating LLMs for developer assistance that addresses severe limitations of prior work. The tasks involved in EvalX are by design realistic and represent several diverse aspects of real-world programming use-cases.
- Authors have designed effective tooling for user preference collection while ensuring several biases are mitigated e.g. top-bottom random swapping; latency masking of the slower model.
- Several ethical considerations have been implemented by the authors given the potentially sensitive nature of data they're collecting from users of this study (Section 3). I'd encourage them to practice utmost care in releasing this dataset.

**Questions For Authors:**

- What would you assess as the primary user motivation to participate? Free generations from commercial LLMs in IDE?

- Do users get to pick the candidate models A and B?

- What does fidelity mean in the context of Fig 2's caption?

- Section 2.3: Any data or citation to support the first sentence?

- Notation could be clearer in Eqn 1 and 2:
    - What does $l$ mean?
    - It is perhaps easier for a reader if you use the notation $l_\text{max}$ instead of using the subscript for $F$ - as the maximum operator is applied on the observed latencies of the 2 models.
    - Did you mean $F$ to be the CDF of latency? $F(x) = P(X < x)$, I don't follow how the CDF of maximum latency would make sense in this context.

- What is the cost incurred so far with the ~11k preferences collected by EvalX or whatever is the total number of completions offered to users?

- When users opt out of data collection - do you still log model comparisons but not their pref/suff/completion code?

- Line 346: How is each element of this matrix defined and what range can it take? I didn't follow how it can $\in R$

**Relation To Broader Scientific Literature:**

Several prior works (Chatbot Arena, LiveBench, BigCodeBench) have focused on evaluation of LLMs in different contexts including for code. The proposed EvalX benchmark attempts to instead capture real world in-IDE use cases when comparing LLMs, which addresses significant gaps in prior evaluation that relied on a fixed set of problems and the LLM's ability to solve these or were only focused on chat based use-cases.

**Theoretical Claims:**

NA.

---

> ### Author Rebuttal · Authors · 2025-04-01
>
> Thank you for the detailed review and appreciation of our work. We aim to address your comments below:
>
> *\[Claim \+ Evidence 1: focus on code completions\]*
>
> * We agree that the paper already provides value to the research community and will be more precise about the scope of our work early on (e.g., we will revise L37 to state “... coding assistants, **specifically focusing on their ability to generate code completions**”).
> * While there are many ways to interact with AI programming assistants, code completions are one of the most frequent use cases, as found by [https://dl.acm.org/doi/10.1145/3597503.3608128](https://dl.acm.org/doi/10.1145/3597503.3608128).
> * Further, EvalX naturally leads to many interesting directions for research for the broader AI and software engineering community, which we discuss in Section 7\. For example, EvalX can be extended to include more interaction modes beyond code completion. Since our submission, we have already added a prompt-based editing feature. The reviewer’s suggestion of long-term impact is another direction that can be studied using data collected from EvalX.
>
> *\[Claim \+ Evidence 2: preferences can be biased, compared to static benchmarks which can be nuanced\]*
>
> * We agree, and believe our work is *complementary* to existing approaches and does not seek to replace them—both are necessary for a holistic view of LLM code evaluation. We discuss this in Related Work (L417), but will update the Introduction to be clearer.
> * We also speculate that building evaluations in the IDE may mitigate the reviewer’s concern for biases towards readability. Since our platform is embedded in a user’s **real development environment**, users are likely to prefer “useful” generations (readability being only one aspect of usefulness) as they would when writing code with Github Copilot.
>
> *\[Experimental design 1: Does the model distribution (skewed towards latency) impact the quality of data? Impact of $\tau$?\]*
>
> * In the analysis conducted in the original submission, we took a few steps to mitigate this impact.
>   * First, when computing Bradley-Terry coefficients, we use an L2 regularization term to prevent overfitting and bias towards models that have received more votes.
>   * Second, we conduct a statistical bootstrap of the leaderboard so we can obtain confidence intervals around the estimated coefficients, improving reliability of our results.
>   * Finally, we ensure that we have reasonable coverage across all model pairs (e.g., $>150$ votes per pair), so votes are not too sparse across some pairs.
> * Inspired by the author's suggestion, we did additional simulations using our data. We upsampled and downsampled model pairs that had fewer and more comparisons respectively so that the difference between the most and least voted pair is within 10%. We find that the rankings remain **identical** to those we report in the original submission. This provides additional confidence that we collected sufficient votes across all pairs.
> * As $\tau$ increases, our coverage becomes more evenly distributed (uniform at $\tau \rightarrow \infty$).
>
> *\[Suggestion 1: More detail about other benchmarks\]*
>
> While we briefly cover metrics for each benchmark in Table 1, we will additionally provide a short summary of each benchmark in Section 4.2.
>
> *\[Suggestion 2: “insights/takeaways for model training or prompting\]*
>
> Since the final version allows an additional page, we will add a new Section 5.3, which will explicitly summarize the insights and takeaways for researchers building new coding assistants. Due to character limits in the response, please see our first response to Reviewer 3 (**1UGk**) for additional details. Additionally, if the reviewer has any suggestions, we would be more than happy to incorporate them\!
>
> *Questions* (not repeated due to space):
>
> * We believe there are three main motivations: 1\) free generations, 2\) overall interest in participating in ML research, and 3\) GitHub Copilot or alternatives may be unavailable in some countries.
> * Models are randomly selected (as discussed in L149).
> * Fidelity refers to the quality of the response, largely in terms of formatting. We will clarify the caption.
> * We observe this in our own data\! Figure 6 shows that 65% of votes are on FiM tasks.
> * l is the maximum latency between models $i$ and $j$ at one point. $F\_{max}(l; i,j)$ is the CDF of these latencies. $F\_{max}(l;i,j) \= P(max(L\_i,L\_j) ≤ l)$ where $L\_i$ and $L\_j$ are random variables for the latency of $i$ and $j$. We agree that $l\_{max}$ is a better notation and will fix this.
> * The total cost is approximately $5k USD.
> * Correct, we always log model comparisons. Users are required to accept this before using the platform.
> * Each element of the matrix is the win-rate between two models over all their battles. The win-rate would be $\in \[0, 1\]$. We will clarify this in the paper by saying $W ∈ ℝ^{(M×M)}$ with $W\_{ij} ∈ \[0,1\] \forall i,j$

---

### Decision · Program_Chairs · 2025-05-01

**Decision:**

Accept (poster)

**Comment:**

This paper introduces a new platform (EvalX) for comparing coding assistants. The EvalX platforms has collected 11K pairwise comparisons from 10 models by integrating an interface into the VSCode IDE, and it proposes a sampling strategy to reduce latency for practical use. This work is undoubtably a great contribution to the research community as understanding the preferences from users in a production environment is crucial in developing next generation of programming tools. Also the ethical considerations that are implemented by the authors are also potentially impactful for follow-up works in collecting user data for coding tasks. Concerns are raise on limited scope, as only code completion is assessed, but code completion is also the majority use case of coding assistants, thus the concern is relatively minor. But I would suggest the authors to add more discussions on the impact on soundness resulted from the skewed distribution of the programming languages, as well as main insights and takeaways derived from the evaluations for the selection of coding assistants for the users, as well as for the model developers.